# `SLowcal-SGD`: Slow Query Points Improve Local-SGD for Stochastic Convex Optimization

**Tehila Dahan**
Department of Electrical Engineering
Technion
Haifa, Israel
t.dahan@campus.technion.ac.il

**Kfir Y. Levy**
Department of Electrical Engineering
Technion
Haifa, Israel
kfirylevy@technion.ac.il

## Abstract

We consider distributed learning scenarios where $M$ machines interact with a parameter server along several communication rounds in order to minimize a joint objective function. Focusing on the heterogeneous case, where different machines may draw samples from different data-distributions, we design the first local update method that provably benefits over the two most prominent distributed baselines: namely Minibatch-SGD and Local-SGD. Key to our approach is a slow querying technique that we customize to the distributed setting, which in turn enables a better mitigation of the bias caused by local updates.

## 1 Introduction

Federated Learning (FL) is a framework that enables huge scale collaborative learning among a large number of heterogeneous[1] clients (or machines). FL may potentially promote fairness among participants, by allowing clients with small scale datasets to participate in the learning process and affect the resulting model. Additionally, participants are not required to directly share data, which may improve privacy. Due to these reasons, FL has gained popularity in the past years, and found use in applications like voice recognition [1, 4], fraud detection [2], drug discovery [3], and more [34].

The two most prominent algorithmic approaches towards federated learning are `Minibatch-SGD` [10] and `Local-SGD` (a.k.a. Federated-Averaging) [27, 28, 33] . In `Minibatch-SGD` all machines (or clients) always compute unbiased gradient estimates of the same query points, while using large batch sizes; and it is well known that this approach is not degraded due to data heterogeneity [37]. On the downside, the number of model updates made by `Minibatch-SGD` may be considerably smaller compared to the number of gradient queries made by each machine; which is due to the use of minibatches. This suggests that there may be room to improve over this approach by employing local update methods like `Local-SGD`, where the number of model updates and the number of gradient queries are the same. And indeed, in the past years, local update methods have been extensively investigated, see e.g. [19] and references therein.

We can roughly divide the research on FL into two scenarios: the *homogeneous* case, where it is assumed that the data on each machine is drawn from the same distribution; and to the more realistic *heterogeneous* case where it is assumed that data distributions may vary between machines.

For the *homogeneous* case it was shown in [36, 13] that the standard `Local-SGD` method is not superior to `Minibatch-SGD`. Nevertheless, [39] have designed an accelerated variant of `Local-SGD` that provably benefits over the Minibatch baseline. These results are established for the fundamental Stochastic Convex Optimization (SCO) setting, which assumes that the learning objective is convex.

---

[1]Heterogeneous here refers to the data of each client, and we assume that its statistical properties may vary between different clients.

Table 1: We compare the best known guarantees for parallel learning, to our SLowcal-SGD approach for the heterogeneous SCO case. The bolded term in the **Rate** column is the one that compares least favourably against Minibatch SGD. Where $G$ and $G_*$ relate to the dissimilarity measures that are defined in Equations (1) and (3). The $\mathbf{R}_{\min}$ column presents the minimal number of communication rounds that are required to obtain a linear speedup (we fixed values of $K$, $M$ and take $\sigma = 1$). Note that we omit methods that do not enable a wall-clock linear speedup with $M$, e.g. [29, 30].

| Method | Rate | $\mathbf{R}_{\min}$ ($\sigma = \mathbf{1}$) |
|---|---|---|
| MiniBatch SGD [10] | $\frac{1}{R} + \frac{\sigma}{\sqrt{MKR}}$ | $MK$ |
| Accelerated MiniBatch SGD [10, 25] | $\frac{1}{R^2} + \frac{\sigma}{\sqrt{MKR}}$ | $(MK)^{1/3}$ |
| Local SGD [37] | $\frac{\mathbf{G^{2/3}}}{\mathbf{R^{2/3}}} + \frac{\sigma^{2/3}}{(\sqrt{K}R)^{2/3}} + \frac{1}{KR} + \frac{\sigma}{\sqrt{MKR}}$ | $G^4 \cdot (MK)^3 + M^3K$ |
| SCAFFOLD [21] | $\frac{\mathbf{1}}{\mathbf{R}} + \frac{\sigma}{\sqrt{MKR}}$ | $MK$ |
| SLowcal-SGD (This paper) | $\frac{\sigma^{1/2} + \mathbf{G_*^{1/2}}}{\mathbf{K^{1/4}R}} + \frac{1}{KR} + \frac{1}{K^{1/3}R^{4/3}} + \frac{1}{R^2} + \frac{\sigma}{\sqrt{MKR}}$ | $G_* \cdot MK^{1/2} + MK^{1/2}$ |
| **Lower Bound:** Local-SGD [39] | $\frac{\mathbf{G_*^{2/3}}}{\mathbf{R^{2/3}}} + \frac{\sigma_*^{2/3}}{(\sqrt{K}R)^{2/3}} + \frac{1}{KR} + \frac{\sigma}{\sqrt{MKR}}$ | $G_*^4 \cdot (MK)^3 + M^3K$ |

Similarly to the homogeneous case, it was shown in [37, 13] that `Local-SGD` is not superior to `Minibatch-SGD` in *heterogeneous* scenarios. Nevertheless, several local approaches that compare with the Minibatch baseline were designed in [21, 14]. Unfortunately, we have so far been missing a local method that provably benefits over the Minibatch baseline in the heterogeneous SCO setting.

Our work focuses on the latter heterogeneous SCO setting, and provide a new `Local-SGD`-style algorithm that provably benefits over the minibatch baseline. Our algorithm named SLowcal-SGD, builds on customizing a recent technique for incorporating a *slowly-changing sequence of query points* [9, 22], which in turn enables to better mitigate the bias induced by the local updates. Curiously, we also found importance weighting to be crucial in order to surpass the minibatch baseline.

In Table 1 we compare our results to the state-of-the-art methods for the heterogeneous SCO setting. We denote $M$ to be the number of machines, $K$ is the number of local updates per round, and $R$ is the number of communications rounds. Additionally, $G$ (or $G_*$) measures the dissimilarity between machines. Our table shows that `Local-SGD` requires much more communication rounds compared to `Minibatch-SGD`, and that the dissimilarity $G$ (or $G^*$) substantially degrades its performance. Conversely, one can see that even if the dissimilarity measure is $G_* = O(1)$, our approach SLowcal-SGD still requires less communication rounds compared to `Minibatch-SGD`.

Similarly to the homogeneous case, accelerated-`Minibatch-SGD` [10, 25], obtains the best performance among all current methods, and it is still open to understand whether one can outperform this accelerated minibatch baseline. In App. A we elaborate on the computations of $R_{\min}$ in Table 1.

**Related Work.** We focus here on centralized learning problems, where we aim to employ $M$ machines in order to minimize a joint learning objective. We allow the machines to synchronize during $R$ communication rounds through a central machine called the *Parameter Server ($\mathcal{PS}$)*; and allow each machine to draw $K$ samples and perform $K$ local gradient computations in every such communication round. We assume that each machine $i$ may draw i.i.d. samples from a distribution $\mathcal{D}_i$, which may vary between machines.

The most natural approach in this context is `Minibatch-SGD`, and its accelerate variant [10], which have been widely adopted both in academy and in industry, see e.g. [16, 32, 38]. Local update methods like `Local-SGD` [28], have recently gained much popularity due to the rise of FL, and have been extensively explored in the past years.

Focusing on the SCO setting, it is well known that the standard `Local-SGD` is not superior (actually in most regimes it is inferior) to `Minibatch-SGD` [36, 37, 13]. Nevertheless, [39] devised a novel accelerated local approach that provably surpasses the Minibatch baseline in the homogeneous case.

The heterogeneous SCO case has also been extensively investigated, with several original and elegant approaches [24, 23, 21, 37, 30, 14, 29, 31]. Nevertheless, so far we have been missing a local approach that provably benefits over `Minibatch-SGD`. Note that [30, 29] improve the communication

complexity with respect to the condition number of the objective; However their performance does not improve as we increase the number of machines $M$ [2], which is inferior to the minibatch baseline.

The heterogeneous non-convex setting was also extensively explored [21, 20, 14]; and the recent work of [31] has developed a novel algorithm that provably benefits over the minibatch baseline in this case. The latter work also provides a lower bound which demonstrates that their upper bound is almost tight. Finally, for the special case of quadratic loss functions, it was shown in [36] and in [21] that it is possible to surpass the minibatch baseline.

It is important to note that excluding the special case of quadratic losses, there does not exist a local update algorithm that provably benefits over *accelerated*-`Minibatch-SGD` [10]. And the latter applies to both homogeneous and heterogeneous SCO problems.

Our local update algorithm utilizes a recent technique of employing slowly changing query points in SCO problems [9]. The latter has shown to be useful in designing universal accelerated methods [22, 12, 5], as well as in improving asynchronous training methods [6].

## 2   Setting: Parallel Stochastic Optimization

We consider Parallel stochastic optimization problems where the objective $f : \mathbb{R}^d \mapsto \mathbb{R}$ is convex and is of the following form,

$$f(x) := \frac{1}{M} \sum_{i \in [M]} f_i(x) := \frac{1}{M} \sum_{i \in [M]} \mathbf{E}_{z^i \sim \mathcal{D}_i} f_i(x; z^i) .$$

Thus, the objective is an average of $M$ functions $\{f_i : \mathbb{R}^d \mapsto \mathbb{R}\}_{i \in [M]}$, and each such $f_i(\cdot)$ can be written as an expectation over losses $f_i(\cdot, z^i)$ where the $z^i$ are drawn from some distribution $\mathcal{D}_i$ which is unknown to the learner. For ease of notation, in what follows we will not explicitly denote $\mathbf{E}_{z^i \sim \mathcal{D}_i}$ but rather use $\mathbf{E}$ to denote the expectation w.r.t. all randomization.

We assume that there exist $M$ machines (computation units), and that each machine may independently draw samples from the distribution $\mathcal{D}_i$, and can therefore compute unbiased gradient estimates to the gradients of $f_i(\cdot)$. Most commonly, we allow the machines to synchronize during $R$ communication rounds through a central machine called the *Parameter Server (PS)*; and allow each machine to perform $K$ local computations in every such communication round.

We consider first order optimization methods that iteratively employ samples and generate a sequence of query points and eventually output a solution $x_{\text{output}}$. Our performance measure is the expected excess loss, **ExcessLoss** $:= \mathbf{E}[f(x_{\text{output}})] - f(w^*)$ , where the expectation is w.r.t. the randomization of the samples, and $w^*$ is a global minimum of $f(\cdot)$ in $\mathbb{R}^d$, i.e., $w^* \in \arg\min_{x \in \mathbb{R}^d} f(x)$.

More concretely, at every computation step, each machine $i \in [M]$ may draw a fresh sample $z^i \sim \mathcal{D}_i$, and compute a gradient estimate $g$ at a given point $x \in \mathbb{R}^d$ as follows, $g := \nabla f_i(x, z^i)$ . and note that $\mathbf{E}[g|x] = \nabla f_i(x)$, i.e. $g$ is an ubiased estimate of $\nabla f_i(x)$.

**General Parallelization Scheme.**   A general scheme for parallel stochastic optimization is described in Alg. 1. It can be seen that the $PS$ communicates with the machines along $R$ communication rounds. In every round $r \in [R]$ the $PS$ distributes an anchor point $\Theta_r$ which is a starting point for the local computations in that round. Based on $\Theta_r$ each machine performs $K$ local gradient computations based on $K$ i.i.d. draws from $\mathcal{D}_i$, and yields a message $\Phi_r^i$. At the end of round $r$ the $PS$ aggregates the messages from all machines and updates the anchor point $\Theta_{r+1}$. Finally, after the last round, the $PS$ outputs $x_{\text{output}}$, which is computed based on the anchor points $\{\Theta_r\}_{r=1}^R$.

Ideally, one would hope that using $M$ machines in parallel will enable to accelerate the learning process by a factor of $M$. And there exists a rich line of works that have shown that this is indeed possible to some extent, depending on $K, R$, and on the parallelization algorithm.

Next, we describe the two most prominent approaches to first-order Parallel Optimization,

---

[2]This implies that such methods do not obtain a wall-clock speedup as we increase the number of machines $M$.

---

**Algorithm 1** Parallel Stochastic Optimization Template

---

Input: $M$ machines, Parameter Server $\mathcal{PS}$, #Communication rounds $R$, #Local computations $K$, initial point $x_0$

$\mathcal{PS}$ Computes initial anchor point $\Theta_0$ using $x_0$

**for** $r = 0, \ldots, R-1$ **do**

    **Distributing anchor:** $\mathcal{PS}$ distributes anchor $\Theta_r$ to all M machines

    **Local Computations:** Each machine $i \in [M]$ performs $K$ local gradient computations based on $K$ i.i.d. draws from $\mathcal{D}_i$, and yields a message $\Phi_r^i$

    **Aggregation:** $\mathcal{PS}$ aggregates $\{\Phi_r^i\}_{i \in [M]}$ from all machines, and computes a new anchor $\Theta_{r+1}$

**end for**

**output:** $\mathcal{PS}$ computes $x_{\text{output}}$ based on $\{\Theta_r\}_{r=1}^R$

---

**(i) Minibatch SGD:** In terms of Alg. 1, one can describe Minibatch-SGD as an algorithm in which the $\mathcal{PS}$ sends a weight vector $x_r \in \mathbb{R}^d$ in every round as the anchor point $\Theta_r$. Based on that anchor $\Theta_r := x_r$, each machine $i$ computes an unbiased gradient estimate based on $K$ independent samples from $\mathcal{D}_i$, i.e. $g_r^i := \frac{1}{K} \sum_{k=1}^K \nabla f_i(x_r, z_{Kr+k}^i)$, and communicates $g_r^i$ as the message $\Phi_r^i$ to the $\mathcal{PS}$. The latter aggregates the messages $\{\Phi_r^i := g_r^i\}_{i \in [M]}$ and compute the next anchor point $x_{r+1}$,

$$x_{r+1} = x_r - \eta \cdot \frac{1}{M} \sum_{i \in [M]} g_r^i \,,$$

where $\eta > 0$ is the learning rate of the algorithm. The benefit in this approach is that all machines always compute gradient estimates at the same anchor points $\{x_r\}_r$, which highly simplifies its analysis. On the downside, in this approach the number of gradient updates $R$ is smaller compared to the number of stochastic gradient computations made by each machine which is $KR$. This gives the hope that there is room to improve upon Minibatch SGD, by mending this issue.

**(ii) Local SGD:** In terms of Alg. 1, one can describe Local-SGD as an algorithm in which the $\mathcal{PS}$ sends a weight vector $x_{rK} \in \mathbb{R}^d$ in every round $r \in [R]$ as the anchor information $\Theta_r$. Based on the anchor $\Theta_r := x_{rK}$, each machine performs a sequence of local gradient updates based on $K$ independent samples from $\mathcal{D}_i$ as follows, $\forall k \in [K]$,

$$x_{rK+k+1}^i = x_{rK+k}^i - \eta \cdot \nabla f_i(x_{rK+k}^i, z_{rK+k}^i) \,,$$

where for all machines $i \in [M]$ we initialize $x_{rK}^i = x_{rK} := \Theta_r$, and $\eta > 0$ is the learning rate of the algorithm. At the end of round $r$ each machine communicates $x_{(r+1)K}^i$ as the message $\Phi_r^i$ to the $\mathcal{PS}$ and the latter computes the next anchor as follows,

$$\Theta_{r+1} := x_{(r+1)K} = \frac{1}{M} \sum_{i \in [M]} x_{(r+1)K}^i \,.$$

In local SGD the number of gradient steps is equal to the number of stochastic gradient computations made by each machine which is $KR$. The latter suggests that such an approach may potentially surpass Minibatch SGD. Nevertheless, this potential benefit is hindered by the bias that is introduced between different machines during the local updates. And indeed, as we show in Table 1, this approach is inferior to Minibatch SGD in the prevalent case where $\sigma = O(1)$.

**Assumptions.** We assume that $f(\cdot)$ is convex, and that the $f_i(\cdot)$ are smooth i.e. $\exists L > 0$ such,

$$\|\nabla f_i(x) - \nabla f_i(y)\| \leq L\|x - y\| \,, \quad \forall i \in [M] \,, \ \forall x, y \in \mathbb{R}^d$$

We also assume that variance of the gradient estimates is bounded, i.e. that there exists $\sigma > 0$ such,

$$\mathbf{E}\|\nabla f_i(x; z) - \nabla f_i(x)\|^2 \leq \sigma^2 \,, \quad \forall x \in \mathbb{R}^d \,, \ \forall i \in [M] \,.$$

Letting $w^*$ be a global minimum of $f(\cdot)$, we assume there exist $G_* \geq 0$ such that,

$$\frac{1}{M} \sum_{i \in [M]} \|\nabla f_i(w^*)\|^2 \leq G_*^2/2 \,, \quad (G_*\text{-\textbf{Dissimilarity}}) \tag{1}$$

The above assumption together with the smoothness and convexity imply (see App. B) ,

$$\frac{1}{M} \sum_{i \in [M]} \|\nabla f_i(x)\|^2 \leq G_*^2 + 4L(f(x) - f(w^*)) , \quad \forall x \in \mathbb{R}^d \tag{2}$$

A stronger dissimilarity assumption that is often used in the literature is the following,

$$\frac{1}{M} \sum_{i \in [M]} \|\nabla f_i(x) - \nabla f(x)\|^2 \leq G^2/2 , \ \forall x \in \mathbb{R}^d \ \textbf{(G-Dissimilarity)} \tag{3}$$

**Notation:** For $\{y_t\}_t$ we denote $y_{t_1:t_2} := \sum_{\tau=t_1}^{t_2} y_\tau$. For $N \in \mathbb{Z}^+$ we denote $[N] := \{0, \ldots, N-1\}$.

# 3 Our Approach

Section 3.1 describes a basic (single machine) algorithmic template called Anytime-GD. Section 3.2 describes our SLowcal-SGD algorithm, which is a Local-SGD style algorithm in the spirit of Anytime GD. We describe our method in Alg. 2, and state its guarantees in Thm. 2.

## 3.1 Anytime GD

The standard GD algorithm computes a sequence of iterates $\{w_t\}_{t \in [T]}$ and queries the gradients at theses iterates. It was recently shown that one can design a GD-style scheme that computes a sequence of iterates $\{w_t\}_{t \in [T]}$ yet queries the gradients at a *different* sequence $\{x_t\}_{t \in [T]}$ which may be *slowly-changing*, in the sense that $\|x_{t+1} - x_t\|$ may be considerably smaller than $\|w_{t+1} - w_t\|$.

Concretely, the Anytime-GD algorithm [9, 22] that we describe in Equations (4) and (5), employs a learning rate $\eta > 0$ and a sequence of non-negative weights $\{\alpha_t\}_t$. The algorithm maintains two sequences $\{w_t\}_t, \{x_t\}_t$ that are updated as follows $\forall t$,

$$w_{t+1} = w_t - \eta \alpha_t g_t , \forall t \in [T] , \text{where } g_t = \nabla f(x_t) , \tag{4}$$

and then,

$$x_{t+1} = \frac{\alpha_{0:t}}{\alpha_{0:t+1}} x_t + \frac{\alpha_{t+1}}{\alpha_{0:t+1}} w_{t+1} . \tag{5}$$

It can be shown that the above implies that $x_{t+1} = \frac{1}{\alpha_{0:t+1}} \sum_{\tau=0}^{t+1} \alpha_\tau w_\tau$, i.e. the $x_t$'s are weighted averages of the $w_t$'s. Thus, at every iterate the gradient $g_t$ is queried at $x_t$ which is a weighted average of past iterates, and then $w_{t+1}$ is updated similarly to GD with a weight $\alpha_t$ on the gradient $g_t$. Moreover, at initialization $x_0 = w_0$.

Curiously, it was shown in [9] that Anytime-GD obtains the same convergence rates as GD for convex loss functions (both smooth and non-smooth). It was further shown and that one can employ a stochastic version (Anytime-SGD) where we query noisy gradients at $x_t$ instead of the exact ones, and that approach performs similarly to SGD.

**Slowly changing query points.** A recent work [6], demonstrates that if we use projected Anytime-SGD, i.e. project the $w_t$ sequence to a given bounded convex domain; then one can immediately show that for both $\alpha_t = 1$ and $\alpha_t = t + 1$ we obtain $\|x_{t+1} - x_t\| \leq 2D/t$, where $D$ is the diameter of the convex domain. Conversely, for standard SGD we have $\|w_{t+1} - w_t\| \leq \eta \|g_t\|$, where $g_t$ here is a (possibly noisy) unbiased estimate of $\nabla f(w_t)$. Thus, while the change between consecutive SGD queries is controlled by $\eta$ which is usually $\propto 1/\sqrt{t}$, and by magnitude of stochastic gradients; for Anytime-SGD the change decays with time, irrespective of the learning rate $\eta$. In [6], this is used to design better and more robust asynchronous training methods.

**Relation to Momentum.** In the appendix we show that Anytime-SGD can be explicitly written as a momentum method, and therefore is quite different from standard SGD. Concretely, for $\alpha_t = 1$ we show that $x_{t+1} \approx x_t - \eta \sum_{\tau=1}^{t} (\tau/t^2) \cdot g_\tau$, and for $\alpha_t \propto t$ we show that $x_{t+1} \approx x_t - \eta \sum_{\tau=1}^{t} (\tau/t)^3 \cdot g_\tau$. Where $g_\tau$ here is a (possibly noisy) unbiased estimate of $\nabla f(x_\tau)$. This momentum interpretation provides a complementary intuition regarding the benefit of Anytime-SGD in the context of local update methods. Momentum brings more stability to the optimization process which in turn reduces the bias between different machines.

For the sake of this paper we will require a specific theorem that does not necessarily regard Anytime-GD, but is rather more general. We will require the following definition,

---

**Algorithm 2** SLowcal-SGD

---

Input: $M$ machines, Parameter Server $\mathcal{PS}$, #Communication rounds $R$, #Local computations $K$, initial point $x_0$, learning rate $\eta > 0$, weights $\{\alpha_t\}_t$

**Initialize:** set $w_0 = x_0$, initialize anchor point $\Theta_0 := (w_0, x_0)$, and set $t = 0$

**for** $r = 0, \ldots, R-1$ **do**

    *Distributing anchor:* $\mathcal{PS}$ distributes anchor $\Theta_r := (w_t, x_t)$ to all machines, each machine $i \in [M]$ initializes $(w_t^i, x_t^i) = \Theta_r := (w_t, x_t)$

    **for** $k = 0, \ldots, K-1$ **do**

        Set $t = rK + k$

        Every machine $i \in [M]$ draws a fresh sample $z_t^i \sim \mathcal{D}_i$, and computes $g_t^i = \nabla f_i(x_t^i, z_t^i)$

        Update $w_{t+1}^i = w_t^i - \eta\alpha_t g_t^i$, and $x_{t+1}^i = \left(1 - \frac{\alpha_{t+1}}{\alpha_{0:t+1}}\right)x_t^i + \frac{\alpha_{t+1}}{\alpha_{0:t+1}}w_{t+1}^i$

    **end for**

    *Aggregation:* $\mathcal{PS}$ aggregates $\{(w_{t+1}^i, x_{t+1}^i)\}_{i\in[M]}$ from all machines, and computes a new anchor $\Theta_{r+1} := (w_{t+1}, x_{t+1}) = \left(\frac{1}{M}\sum_{i\in[M]}w_{t+1}^i, \frac{1}{M}\sum_{i\in[M]}x_{t+1}^i\right)$

**end for**

**output:** $\mathcal{PS}$ outputs $x_T$ (recall $T = KR$)

---

**Definition** *Let $\{\alpha_t \geq 0\}_t$ be a sequence of non-negative weights, and let $\{w_t \in \mathbb{R}^d\}_t$, be an arbitrary sequence. We say that a sequence $\{x_t \in \mathbb{R}^d\}_t$ is an $\{\alpha_t\}_t$ weighted average of $\{w_t\}_t$ if $x_0 = w_0$, and for any $t > 0$ Eq. (5) is satisfied.*

Next, we state the main theorem for this section, which applies for any sequence $\{w_t \in \mathbb{R}^d\}_t$,

**Theorem 1** (Rephrased from Theorem 1 in [9])**.** *Let $f : \mathbb{R}^d \mapsto \mathbb{R}$ be a convex function with a global minimum $w^*$. Also let $\{\alpha_t \geq 0\}_t$, and $\{w_t \in \mathbb{R}^d\}_t$, $\{x_t \in \mathbb{R}^d\}_t$ such that $\{x_t\}_t$ is an $\{\alpha_t\}_t$ weighted average of $\{w_t\}_t$. Then the following holds for any $t \geq 0$,*

$$0 \leq \alpha_{0:t}\left(f(x_t) - f(w^*)\right) \leq \sum_{\tau=0}^{t} \alpha_\tau \nabla f(x_\tau) \cdot (w_\tau - w^*)\,.$$

### 3.2 SLowcal-SGD

Our approach is to employ an Anytime version of Local-SGD, which we name by SLowcal-SGD.

**Notation:** Prior to describing our algorithm we will define $t$ to be the total of *per-machine* local updates up to step $k$ of round $r$, resulting $t := rK + k$. In what follows, we will often find it useful to denote the iterates and samples using $t$, rather than explicitly denoting $t = rK + k$. Additionally we use $\{\alpha_t\}_t$ to denote a pre-defined sequence of non-negative weights. Finally, we denote $T := RK$.

In the spirit of Anytime-SGD our approach is to maintain two sequences per machine $i \in [M]$: $\{w_t^i \in \mathbb{R}^d\}_t$ and $\{x_t^i \in \mathbb{R}^d\}_t$. Our approach is depicted explicitly in Alg. 2. Next we describe our algorithm in terms of the scheme depicted in Alg. 1:

**(i) Distributing anchor.** At the beginning of round $r$ the $\mathcal{PS}$ distributes $\Theta_r = (w_t, x_t) = (w_{rK}, x_{rK}) \in \mathbb{R}^d \times \mathbb{R}^d$ to all machines.

**(ii) Local Computations.** For $t = rK$, every machine initializes $(w_t^i, x_t^i) = \Theta_r$, and for the next $K$ rounds, i.e. for any $rK \leq t \leq (r+1)K - 1$, every machine performs a sequence of local Anytime-SGD steps as follows,

$$w_{t+1}^i = w_t^i - \eta\alpha_t g_t^i\,, \tag{6}$$

where similarly to Anytime-SGD we query the gradients at the averages $x_t^i$, meaning $g_t^i = \nabla f_i(x_t^i, z_t^i)$. And query points are updated as weighted averages of past iterates $\{w_t\}_t$, ,

$$x_{t+1}^i = (1 - \frac{\alpha_{t+1}}{\alpha_{0:t+1}})x_t^i + \frac{\alpha_{t+1}}{\alpha_{0:t+1}}w_{t+1}^i\,, \qquad \forall\, rK \leq t \leq (r+1)K - 1\,. \tag{7}$$

At the end round $r$, i.e. $t = (r+1)K$, each machine communicates $(w_t^i, x_t^i)$ as a message to the $\mathcal{PS}$.

**(iii) Aggregation.** The $\mathcal{PS}$ aggregates the messages and computes the next anchor point $\Theta_{r+1} = (w_t, x_t) = \frac{1}{M}\sum_{i\in[M]}\Phi_r^i := \left(\frac{1}{M}\sum_{i\in[M]}w_t^i, \frac{1}{M}\sum_{i\in[M]}x_t^i\right)$, where $t = (r+1)K$.

**Remark:** Note that for $t = rK$ our notation for $(w_t^i, x_t^i)$ is inconsistent: at the end of round $r - 1$ these values may vary between different machines, while at the beginning of round $r$ these values are all equal to $\Theta_r := (w_t, x_t)$. Nevertheless, for simplicity we will abuse notation, and explicitly state the right definition when needed. Importantly, in most of our analysis we will mainly need to refer to the averages $\left( \frac{1}{M} \sum_{i \in [M]} w_t^i, \frac{1}{M} \sum_{i \in [M]} x_t^i \right)$, and note the latter are consistent at the end and beginning of consecutive rounds due to the definition of $\Theta_r$, and $\Phi_{r-1}^i$.

### 3.2.1 Guarantees & Intuition

Below we state our main result for SLowcal-SGD (Alg. 2),

**Theorem 2.** *Let $f(\cdot)$ be a convex and $L$-smooth function. Then under the assumption that we make in Sec. 2, invoking Alg. 2 with weights $\{\alpha_t = t + 1\}_{t \in [T]}$, and an appropriate learning rate $\eta$ ensures,*

$$\mathbf{E}\Delta_T \leq O\left( LB_0^2 \left( \frac{1}{KR} + \frac{1}{R^2} + \frac{1}{K^{1/3}R^{4/3}} \right) + \frac{\sigma B_0}{\sqrt{MKR}} + \frac{L^{1/2}(\sigma^{1/2} + G_*^{1/2}) \cdot B_0^{3/2}}{K^{1/4}R} \right),$$

*where $\Delta_T := f(x_T) - f(x^*)$, $B_0 := \|w_0 - w^*\|$, and we choose the learning rate as follows,*

$$\eta = \min\left\{ \frac{1}{48L(T+1)}, \frac{1}{10LK^2}, \frac{1}{40LK(T+1)^{2/3}}, \frac{\|w_0 - w^*\|\sqrt{M}}{\sigma T^{3/2}}, \frac{\|w_0 - w^*\|^{1/2}}{L^{1/2}K^{7/4}R(\sigma^{1/2} + G_*^{1/2})} \right\} \quad (8)$$

As Table 1 shows, Thm. 2 implies that SLowcal-SGD improves over all existing upper bounds for Minibatch and Local SGD, by allowing less communication rounds to obtain a linear speedup of $M$.

**Intuition.** The degradation in local SGD schemes (both standard and Anytime) is due to the bias that it introduces between different machines during each round, which leads to a bias in their gradients. Intuitively, this bias is small if the machines query the gradients at a sequence of slowly changing query points. This is exactly the benefit of SLowcal-SGD which queries the gradients at averaged iterates $x_t^i$'s. Intuitively these averages are slowly changing compared to the iterates themselves $w_t^i$; and recall that the latter are the query points used by standard Local-SGD. A complementary intuition to the benefit of our approach, is the interpretation of Anytime-SGD as a momentum method (see Sec. 3.1 and the appendix) which leads to decreased bias between machines.

To further simplify the more technical discussion here, we will assume the homogeneous case, i.e., that for any $i \in [M]$ we have $\mathcal{D}_i = \mathcal{D}$ and $f_i(\cdot) = f(\cdot)$.

So a bit more formally, let us discuss the bias between query points in a given round $r \in [R]$, and let us denote $t_0 = rK$. The following holds for standard **Local SGD**,

$$w_t^i = w_{t_0} - \eta \sum_{\tau=t_0}^{t-1} g_\tau^i, \quad \forall i \in [M], t \in [t_0, t_0 + K]. \quad (9)$$

where $g_\tau^i$ is the noisy gradients that Machine $i$ computes in $w_\tau^i$, and we can write $g_\tau^i := \nabla f(w_\tau^i) + \xi_\tau^i$, where $\xi_\tau^i$ is the noisy component of the gradient. Thus, for two machines $i \neq j$ we can write,

$$\mathbf{E}\|w_t^i - w_t^j\|^2 = \eta^2 \mathbf{E}\left\| \sum_{\tau=t_0}^{t-1} g_\tau^i - g_\tau^j \right\|^2 \approx \eta^2 \mathbf{E}\left\| \sum_{\tau=t_0}^{t-1} \nabla f(w_\tau^i) - \nabla f(w_\tau^j) \right\|^2 + \eta^2 \mathbf{E}\left\| \sum_{\tau=t_0}^{t-1} \xi_\tau^i - \xi_\tau^j \right\|^2$$

And it was shown in [36], that the noisy term is dominant and therefore we can bound,

$$\frac{1}{\eta^2} \mathbf{E}\|w_t^i - w_t^j\|^2 \lesssim \mathbf{E}\left\| \sum_{\tau=t_0}^{t-1} \xi_\tau^i - \xi_\tau^j \right\|^2 \approx t - t_0 \leq K. \quad (10)$$

Similarly, for **SLowcal-SGD** we would like to bound $\mathbf{E}\|x_t^i - x_t^j\|^2$ for two machines $i \neq j$; and in order to simplify the discussion we will assume uniform weights i.e., $\alpha_t = 1$, $\forall t \in [T]$. Now the update rule for the iterates $w_t^i$, is of the same form as in Eq. (9), only now $g_\tau^i := \nabla f(x_\tau^i) + \xi_\tau^i$, where $\xi_\tau^i$ is the noisy component of the gradient. Consequently,

$$\sum_{\tau=t_0}^{t} (w_\tau^i - w_\tau^j) \approx -\eta \sum_{\tau=t_0}^{t-1} (t - \tau)(g_\tau^i - g_\tau^j) \approx -\eta K \sum_{\tau=t_0}^{t-1} (g_\tau^i - g_\tau^j),$$

where we took a crude approximation of $t - \tau \approx K$. Now, by definition of $x_t^i$ and $\alpha_t = 1$, $x_t^i = \frac{t_0}{t} \cdot x_{t_0} + \frac{1}{t} \sum_{\tau=t_0}^{t} w_\tau^i$, $\forall i \in [M], t \in [t_0, t_0 + K]$. Thus, for two machines $i \neq j$ we have,

$$\frac{1}{\eta^2}\mathbf{E}\|x_t^i - x_t^j\|^2 = \frac{1}{\eta^2}\mathbf{E}\left\|\frac{1}{t}\sum_{\tau=t_0}^{t} w_\tau^i - w_\tau^j\right\|^2 \approx \frac{1}{\eta^2} \cdot \frac{\eta^2 K^2}{t^2}\mathbf{E}\left\|\sum_{\tau=t_0}^{t-1} g_\tau^i - g_\tau^j\right\|^2$$

$$\approx \frac{K^2}{t^2}\mathbf{E}\left\|\sum_{\tau=t_0}^{t-1} \nabla f(x_\tau^i) - \nabla f(x_\tau^j)\right\|^2 + \frac{K^2}{t^2}\mathbf{E}\left\|\sum_{\tau=t_0}^{t-1} \xi_\tau^i - \xi_\tau^j\right\|^2 .$$

As we show in our analysis, the noisy term is dominant, so we can therefore bound,

$$\frac{1}{\eta^2}\mathbf{E}\|x_t^i - x_t^j\|^2 \lesssim \frac{K^2}{t^2}\mathbf{E}\left\|\sum_{\tau=t_0}^{t-1} \xi_\tau^i - \xi_\tau^j\right\|^2 \approx \frac{K^2(t - t_0)}{t^2} \leq \frac{K^3}{t^2} . \tag{11}$$

Taking $t \approx T = RK$ above yields a bound of $O(K/R^2)$. Thus Equations (10), (11), illustrate that the bias of SLowcal-SGD is smaller by a factor of $R^2$ compared to the bias of standard Local-SGD. In the appendix we demonstrate the same benefit of Anytime-SGD over SGD when both use $\alpha_t \propto t$.

Finally, note that the biases introduced by the local updates come into play in a slightly different manner in Local-SGD compared to SLowcal-SGD [3]. Consequently, the above discussion does not enable to demonstrate the exact rates that we derive. Nevertheless, it provides some intuition regarding the benefit of our approach. The full and exact derivations appear in the appendix.

**Importance Weights.** One may wonder whether it is necessary to employ increasing weights $\alpha_t = t+1$, rather than employing standard uniform weights $\alpha_t = 1$, $\forall t$. Surprisingly, in our analysis we have found that increasing weights are crucial in order to obtain a benefit over Minibatch-SGD, and that upon using uniform weights SLowcal-SGD performs worse compared to Minibatch SGD! We elaborate on this in Appendix L. Below we provide an intuitive explanation.

**Intuitive Explanation.** The intuition behind the importance of using increasing weights is the following: Increasing weights are a technical tool to put more emphasis on the last rounds. Now, in the context of Local update methods, the iterates of the last rounds are more attractive since the bias between different machines shrinks as we progress. Intuitively, this happens since as we progress with the optimization process, the expected value of the gradients that we compute goes to zero (since we converge); and consequently the bias between different machines shrinks as we progress.

### 3.3 Proof Sketch for Theorem 2

*Proof Sketch for Theorem 2.* As a starting point for the analysis, for every iteration $t \in [T]$ we will define the averages of $(w_t^i, x_t^i, g_t^i)$ across all machines as follows,

$$w_t := \frac{1}{M}\sum_{i\in[M]} w_t^i , \quad \& \quad x_t := \frac{1}{M}\sum_{i\in[M]} x_t^i \quad \& \quad g_t := \frac{1}{M}\sum_{i\in[M]} g_t^i .$$

Note that Alg. 2 explicitly computes $(w_t, x_t)$ only once every $K$ local updates, and that theses are identical to the local copies of every machine at the beginning of every round. Combining the above definitions with Eq. (6) yields,

$$w_{t+1} = w_t - \eta\alpha_t g_t , \ \forall t \in [T] \tag{12}$$

Further combining these definitions with Eq. (7) yields,

$$x_{t+1} = (1 - \frac{\alpha_{t+1}}{\alpha_{0:t+1}})x_t + \frac{\alpha_{t+1}}{\alpha_{0:t+1}}w_{t+1} , \ \forall t \in [T] \tag{13}$$

The above implies that the $\{x_t\}_{t\in[T]}$ sequence is an $\{\alpha_t\}_{t\in[T]}$ weighted average of $\{w_t\}_{t\in[T]}$. This enables to employ Thm. 1 which yields, $\alpha_{0:t}\Delta_t \leq \sum_{\tau=0}^{t} \alpha_\tau \nabla f(x_\tau) \cdot (w_\tau - w^*)$ , where we denote $\Delta_t := f(x_t) - f(w^*)$. This bound highlights the challenge in the analysis: our algorithm does not directly compute unbiased estimates of $x_t$, except for the first iterate of each round. Concretely, Eq. (12) implies that our algorithm effectively updates using $g_t$ which is a biased estimate of $\nabla f(x_t)$.

---

[3]A major challenge in our analysis is that for a given $i \in [M]$ the $\{x_t^i\}_t$ sequence is **not** necessarily an $\{\alpha_t\}_t$ weighted average of the $\{w_t^i\}_t$.

It is therefore natural to decompose $\nabla f(x_\tau) = g_\tau + (\nabla f(x_\tau) - g_\tau)$ in the above bound, yielding,

$$\alpha_{0:t}\Delta_t \leq \underbrace{\sum_{\tau=0}^{t} \alpha_\tau g_\tau \cdot (w_\tau - w^*)}_{(A)} + \underbrace{\sum_{\tau=0}^{t} \alpha_\tau (\nabla f(x_\tau) - g_\tau) \cdot (w_\tau - w^*)}_{(B)} \tag{14}$$

Thus, we intend to bound the weighted error $\alpha_{0:t}\Delta_t$ by bounding two terms: (A) which is related to the update rule of the algorithm, and (B) which accounts for the bias between $g_t$ and $\nabla f(x_t)$.

**Notation:** In what follows we will find the following notation useful, $\bar{g}_t := \frac{1}{M}\sum_{i\in[M]}\nabla f_i(x_t^i)$, and note that $\bar{g}_t = \mathbf{E}\left[g_t|\{x_t^i\}_{i\in[M]}\right]$. We will also employ the following notations: $V_t := \sum_{\tau=0}^{t}\alpha_\tau^2\|\bar{g}_\tau - \nabla f(x_\tau)\|^2$, and $D_t := \|w_t - w^*\|^2$, where $w^*$ is a global minimum of $f(\cdot)$. We will also denote $D_{0:t} := \sum_{\tau=0}^{t}\|w_\tau - w^*\|^2$.

**Bounding (A):** Due to the update rule of Eq. (12), one can show by standard regret analysis that:
(A) $:= \sum_{\tau=0}^{t}\alpha_\tau g_\tau \cdot (w_\tau - w^*) \leq \frac{\|w_0-w^*\|^2}{2\eta} + \frac{\eta}{2}\sum_{\tau=0}^{t}\alpha_\tau^2\|g_\tau\|^2$ ,

**Bounding (B):** We can bound (B) in expectation using $V_t$ and $D_{0:t}$ as follows for any $\rho > 0$:
$\mathbf{E}\left[(B)\right] \leq \frac{1}{2\rho}\mathbf{E}V_t + \frac{\rho}{2}\mathbf{E}D_{0:t}$ ,

**Combining (A) and (B):** Combining the above bounds on (A) and (B) into Eq. (14) we obtain the following bound which holds for any $\rho > 0$ and $t \in [T]$,

$$\alpha_{0:t}\mathbf{E}\Delta_t \leq \frac{\|w_0 - w^*\|^2}{2\eta} + \frac{\eta}{2}\mathbf{E}\sum_{\tau=0}^{T}\alpha_\tau^2\|g_\tau\|^2 + \frac{1}{2\rho}\mathbf{E}V_T + \frac{\rho}{2}\mathbf{E}D_{0:T} \tag{15}$$

Now, to simplify the proof sketch we shall assume that $D_t \leq D_0 \ \forall t$, implying that $D_{0:T} \leq TD_0$. Plugging this into the above equation and taking $\rho = \frac{1}{4\eta T}$ gives,

$$\alpha_{0:t}\mathbf{E}\Delta_t \leq \frac{\|w_0 - w^*\|^2}{\eta} + \eta \cdot \underbrace{\mathbf{E}\sum_{\tau=0}^{T}\alpha_\tau^2\|g_\tau\|^2}_{(*)} + 4\eta T\mathbf{E}V_T . \tag{16}$$

Next we will bound $(*)$ and $\mathbf{E}V_T$, and plug them back into Eq. (16).

**Bounding $(*)$:** To bound $(*)$ it is natural to decompose $g_\tau = (g_\tau - \bar{g}_\tau) + (\bar{g}_\tau - \nabla f(x_\tau)) + \nabla f(x_\tau)$. Using this decomposition we show that, $(*) \lesssim 3\frac{\sigma^2}{M}\sum_{t=0}^{T}\alpha_t^2 + 3\mathbf{E}V_T + 12L\mathbf{E}\sum_{t=0}^{T}\alpha_{0:t}\Delta_t$ .

**Bounding $\mathbf{E}V_T$** The definition of $V_t$ shows that it is encompasses the bias that is introduced due to the local updates, which in turn relates to the distances $\|x_t^i - x_t^j\|$ , $\forall i,j \in [M]$. Thus, $\mathbf{E}V_T$ is therefore directly related to the dissimilarity between the machines. Our analysis shows the following: $\mathbf{E}V_T \leq 400L^2\eta^2K^3\sum_{\tau=0}^{T}\alpha_{0:\tau}\cdot(G_*^2 + 4L\Delta_\tau) + 90L^2\eta^2K^6R^3\sigma^2$ . Plugging the above into Eq. (16), and using our choice for $\eta$, gives an almost explicit bound,

$$\alpha_{0:t}\mathbf{E}\Delta_t \lesssim \frac{\|w_0 - w^*\|^2}{\eta} + \eta\frac{\sigma^2}{M}\sum_{t=0}^{T}\alpha_t^2 + L^2\eta^3TK^6R^3\sigma^2 + L^2\eta^3TK^3\sum_{\tau=0}^{T}\alpha_{0:\tau}G_*^2 + \frac{1}{2(T+1)}\mathbf{E}\sum_{t=0}^{T}\alpha_{0:t}\Delta_t .$$

The theorem follows by plugging above the choices of $\eta, \alpha_t$, and using a technical lemma. $\qquad\square$

## 4  Experiments

To assess the effectiveness of our proposed approach, we conducted experiments on the MNIST [26] dataset—a well-established benchmark in image classification comprising 70,000 grayscale images of handwritten digits (0–9), with 60,000 images designated for training and 10,000 for testing. The dataset was accessed via `torchvision` (version 0.16.2). We implemented a logistic regression

model [7] using the PyTorch framework and executed all computations on an NVIDIA L40S GPU. To ensure robustness, results were averaged over three different random seeds. The complete codebase for these experiments is publicly available on our GitHub repository.[4]

We evaluated our approach using parameters derived from our theoretical framework ($\alpha_t = t$) in comparison to Local-SGD and Minibatch-SGD under various configurations. Specifically, experiments were conducted with 16, 32, and 64 workers to examine the scalability and robustness of the proposed method. We also varied the number of local updates $K$ (or minibatch sizes for Minibatch-SGD) among 4, 8, 16, 32, and 64 to investigate how different local iteration counts impact performance. Data subsets for each worker were generated using a Dirichlet distribution [17] with $\alpha = 0.1$ to simulate real-world non-IID data scenarios characterized by high heterogeneity. For fairness, the learning rate was selected through grid search, with a value of 0.01 for SLowcal-SGD and Local-SGD, and 0.1 for Minibatch-SGD. More details about the data distribution across workers and complete experimental results are provided in Appendix M.

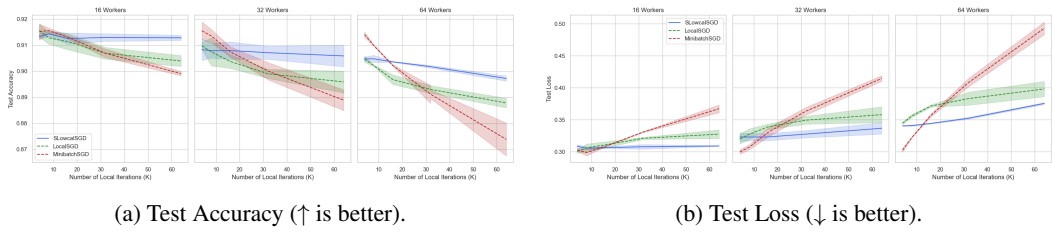

(a) Test Accuracy (↑ is better).          (b) Test Loss (↓ is better).

Figure 1: Performance vs. Local Iterations ($K$) for different numbers of workers ($M$).

Our results on the MNIST dataset, presented in Figure 1 and detailed in Appendix M.2, demonstrate the effectiveness of our approach, showing consistent performance improvements compared to Local-SGD and Minibatch-SGD as the number of local steps increases. Notably, this improvement becomes even more significant compared to the other methods as the number of workers increases, underscoring the scalability of our method and aligning with the theoretical guarantees outlined in our framework. These results highlight the robustness of our approach in handling highly heterogeneous, distributed environments.

Upon closer inspection, when a small number of local steps are performed, the differences between the approaches are negligible, with a slight advantage for Minibatch-SGD. However, as the number of local steps increases, the minibatch size grows, and the need for significant variance reduction diminishes. In this regime, making more frequent optimization updates becomes more impactful, as demonstrated by the superior performance of the local approaches compared to Minibatch-SGD. Importantly, with SLowcal-SGD, which keeps local updates closely aligned among workers throughout the training process, we can achieve significantly better and more stable performance compared to both Minibatch-SGD and Local-SGD as the number of local steps $K$ and the number of workers $M$ increase.

## 5 Conclusion

We have presented the first local approach for the heterogeneous distributed Stochastic Convex Optimization (SCO) setting that provably benefits over the two most prominent baselines, namely Minibatch-SGD, and Local-SGD. There are several interesting avenues for future exploration:
**(a)** developing an adaptive variant that does not require the knowledge of the problem parameters like $\sigma$ and $L$; **(b)** Allowing a per dimension step-size that could benefit in (the prevalent) scenarios where the scale of the gradients considerably changes between different dimensions; in the spirit of the well known AdaGrad method [11]. Finally, **(c)** it will be interesting to understand whether we can find an algorithm that provably dominates over the Accelerated Minibach-SGD baseline, which is an open question also in the homogeneous SCO setting.

---

[4] https://github.com/dahan198/slowcal-sgd

## Acknowledgement

This research was partially supported by Israel PBC-VATAT, the Technion Artificial Intelligent Hub (Tech.AI), and the Israel Science Foundation (grant No. 3109/24).

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

# A  Explanations Regarding the Linear Speedup and Table 1

Here we elaborate on the computations done in Table 1. First we will explain why the dominance of the term $\frac{1}{\sqrt{MKR}}$ implies a linear speedup by a factor of $M$.

**Explanation.**  Recall that using SGD with a single machine $M = 1$, yields a convergnece rate of $\frac{1}{\sqrt{KR}}$ (as a dominant term). Thus, in order to obtain an excess loss smaller than some $\varepsilon > 0$, SGD requires $RK \geq \Omega\left(\frac{1}{\varepsilon^2}\right)$. Where $RK$ is the wall-clock time required to compute the solution.

Now, when we use parallel optimization with $R$ communication rounds, $K$ local computations, and $M$ machines, the wall-clock time to compute a solution is still $RK$. Now, if the dominant term in the convergence rate of this algorithm is $\frac{1}{\sqrt{MKR}}$ then the wall clock time to obtain an $\varepsilon$-optimal solution should be $RK \geq \Omega\left(\frac{1}{M\varepsilon^2}\right)$. And the latter is smaller by a factor of $M$ compared to a single machine.

**Computation of $R_{\min}$ in Table 1.**  The term $\frac{1}{\sqrt{MKR}}$ appears in the bounds of all of the parallel optimization methods that we describe. Nevertheless, it is dominant up as long as the number of communication rounds $R$ is larger than some treshold value $R_{\min}$, that depends on the specific convergence rate. Clearly, smaller values of $R_{\min}$ imply less communication. Thus, in the $\mathbf{R}_{\min}$ column of the table we compute $R_{\min}$ for each method based on the term in the bound that compares least favourably against $\frac{1}{\sqrt{MKR}}$. These terms are bolded in the **Rate** column of the table.

Concretely, denoting this less favourable term by $\mathcal{B}^{\text{parallel}} := \mathcal{B}^{\text{parallel}}(M, K, R, G_*)$ [5], then $\mathbf{R}_{\min}$ is the lowest $R$ which satisfies,

$$\mathcal{B}^{\text{parallel}} \leq \frac{1}{\sqrt{MKR}} \ .$$

# B  On Heterogeneity Assumption

Let us assume that the following holds at the optimum $w^*$,

$$\frac{1}{M} \sum_{i \in [M]} \|\nabla f_i(w^*)\|^2 \leq G_*^2/2$$

Then we can show the following relation for any $w \in \mathbb{R}^d$,

$$\begin{aligned}
\frac{1}{M} \sum_{i \in [M]} \|\nabla f_i(w)\|^2 &= \frac{1}{M} \sum_{i \in [M]} \|\nabla f_i(w) - \nabla f_i(w^*) + \nabla f_i(w^*)\|^2 \\
&\leq \frac{2}{M} \sum_{i \in [M]} \|\nabla f_i(w) - \nabla f_i(w^*)\|^2 + \frac{2}{M} \sum_{i \in [M]} \|\nabla f_i(w^*)\|^2 \\
&\leq 4L(f(w) - f(w^*)) + G_*^2 \ .
\end{aligned}$$

where we used $\|a + b\|^2 \leq 2\|a\|^2 + 2\|b\|^2$ which holds for any $a, b \in \mathbb{R}^d$, and the last line follows by the lemma below that we borrow from [18, 8].

**Lemma 1.** *Let $\mathcal{L}(x) = \frac{1}{M} \sum_{i \in [M]} \ell_i(x)$ be a convex function with global minimum $w^*$, and assume that every $f_i : \mathbb{R}^d \mapsto \mathbb{R}$ is L-smooth. Then the following holds,*

$$\frac{1}{M} \sum_{i \in [M]} \|\nabla \ell_i(w) - \nabla \ell_i(w^*)\|^2 \leq 2L(\mathcal{L}(w) - \mathcal{L}(w^*)) \ .$$

*Proof of Lemma 1.* The lemma follows immediately from lemma 27.1 in [8], by taking $v = w^*$ therein. □

---

[5] $\mathcal{B}^{\text{parallel}}$ may also depend on $\sigma, L, \|w_0 - w^*\|$ but for simplicity of exposition we hide these dependencies in Table 1.

# C   Interpreting Anytime-SGD as Momentum

Here we show how to interpret the Anytime-SGD algorithm that we present in Equations (4),(5), as a momentum method. For completeness we rewrite the update equations below,

$$w_{t+1} = w_t - \eta \alpha_t g_t \ , \forall t \in [T] \ , \text{where } g_t = \nabla f(x_t) \ , \tag{17}$$

and then,

$$x_{t+1} = \frac{\alpha_{0:t}}{\alpha_{0:t+1}} x_t + \frac{\alpha_{t+1}}{\alpha_{0:t+1}} w_{t+1} \ . \tag{18}$$

where $g_t$ is an unbiased gradient estimate at $x_t$, and $\{\alpha_t\}_t$ is a sequence of non-negative scalars. And at initialization $x_0 = w_0$.

First note that Eq. (18) directly implies that,

$$x_{t+1} = \frac{1}{\alpha_{0:t+1}} \sum_{\tau=0}^{t+1} \alpha_\tau w_\tau \ .$$

Next, note that we can directly write,

$$w_\tau = w_0 - \eta \sum_{n=0}^{\tau-1} \alpha_n g_n$$

Plugging the above into the formula for $x_{t+1}$ yields,

$$x_{t+1} = w_0 - \eta \frac{1}{\alpha_{0:t+1}} \sum_{\tau=0}^{t+1} \sum_{n=0}^{\tau-1} \alpha_\tau \alpha_n g_n$$

$$= w_0 - \eta \frac{1}{\alpha_{0:t+1}} \sum_{n=0}^{t} \sum_{\tau=n+1}^{t+1} \alpha_\tau \alpha_n g_n$$

$$= w_0 - \eta \frac{1}{\alpha_{0:t+1}} \sum_{n=0}^{t} \alpha_{n+1:t+1} \alpha_n g_n \ .$$

Thus,

$$\frac{1}{\eta} (x_{t+1} - x_t) = \frac{1}{\alpha_{0:t}} \sum_{n=0}^{t-1} \alpha_{n+1:t} \alpha_n g_n - \frac{1}{\alpha_{0:t+1}} \sum_{n=0}^{t} \alpha_{n+1:t+1} \alpha_n g_n$$

$$= \frac{1}{\alpha_{0:t+1}} \sum_{n=0}^{t-1} \frac{\alpha_{0:t+1}}{\alpha_{0:t}} \alpha_{n+1:t} \alpha_n g_n - \frac{1}{\alpha_{0:t+1}} \sum_{n=0}^{t-1} \alpha_{n+1:t+1} \alpha_n g_n - \frac{1}{\alpha_{0:t+1}} \alpha_{t+1} \alpha_t g_t$$

$$= -\frac{1}{\alpha_{0:t+1}} \sum_{n=0}^{t-1} \left( \alpha_{n+1:t+1} - \frac{\alpha_{0:t+1}}{\alpha_{0:t}} \alpha_{n+1:t} \right) \alpha_n g_n - \frac{1}{\alpha_{0:t+1}} \alpha_{t+1} \alpha_t g_t$$

$$= -\frac{1}{\alpha_{0:t+1}} \sum_{n=0}^{t-1} \alpha_{t+1} \frac{\alpha_{0:n}}{\alpha_{0:t}} \alpha_n g_n - \frac{1}{\alpha_{0:t+1}} \alpha_{t+1} \alpha_t g_t$$

$$= -\frac{1}{\alpha_{0:t+1}} \sum_{n=0}^{t} \alpha_{t+1} \alpha_n \frac{\alpha_{0:n}}{\alpha_{0:t}} g_n \ , \tag{19}$$

where we used the equality below,

$$\alpha_{n+1:t+1} - \frac{\alpha_{0:t+1}}{\alpha_{0:t}} \alpha_{n+1:t} = \alpha_{t+1} - \alpha_{n+1:t} \frac{\alpha_{t+1}}{\alpha_{0:t}} = \alpha_{t+1} \left( 1 - \frac{\alpha_{n+1:t}}{\alpha_{0:t}} \right) = \alpha_{t+1} \frac{\alpha_{0:n}}{\alpha_{0:t}} \ .$$

Thus we can write,

$$x_{t+1} \approx x_t - \eta \frac{1}{\alpha_{0:t+1}} \sum_{n=0}^{t} \alpha_{t+1} \alpha_n \frac{\alpha_{0:n}}{\alpha_{0:t}} g_n \ ,$$

**Uniform Weights.** Thus, taking uniform weights $\alpha_t = 1$ yields,

$$x_{t+1} \approx x_t - \eta \sum_{n=0}^{t} \frac{n}{t^2} g_n \ .$$

**Linear Weights.** Similarly, taking linear weights $\alpha_t = t + 1$ yields,

$$x_{t+1} \approx x_t - \eta \sum_{n=0}^{t} \frac{n^3}{t^3} g_n \ .$$

## D   Proof of Theorem 1

*Proof of Theorem 1.* We rehearse the proof of Theorem 1 from [9].

First, since $w^*$ is a global minimum and $\alpha_{0:t}$ are non-negative than clearly,

$$\alpha_{0:t} \left( f(x_t) - f(w^*) \right) \geq 0 \ .$$

Now, notice that the following holds,

$$\alpha_t (x_t - w_t) = \alpha_{0:t-1}(x_{t-1} - x_t)$$

Using the gradient inequality for $f$ gives,

$$
\begin{aligned}
\sum_{\tau=0}^{t} \alpha_\tau (f(x_\tau) - f(w^*)) &\leq \sum_{\tau=0}^{t} \alpha_\tau \nabla f(x_\tau) \cdot (x_\tau - w^*) \\
&= \sum_{\tau=0}^{t} \alpha_\tau \nabla f(x_\tau) \cdot (w_\tau - w^*) + \sum_{\tau=0}^{t} \alpha_\tau \nabla f(x_\tau) \cdot (x_\tau - w_\tau) \\
&= \sum_{\tau=0}^{t} \alpha_\tau \nabla f(x_\tau) \cdot (w_\tau - w^*) + \sum_{\tau=0}^{t} \alpha_{0:\tau-1} \nabla f(x_\tau) \cdot (x_{\tau-1} - x_\tau) \\
&\leq \sum_{\tau=0}^{t} \alpha_\tau \nabla f(x_\tau) \cdot (w_\tau - w^*) + \sum_{\tau=0}^{t} \alpha_{0:\tau-1}(f(x_{\tau-1}) - f(x_\tau)) \ ,
\end{aligned}
$$

where we have used the gradient inequality again which implies $\nabla f(x_\tau) \cdot (x_{\tau-1} - x_\tau) \leq f(x_{\tau-1}) - f(x_\tau)$.

Now Re-ordering we obtain,

$$\sum_{\tau=0}^{t} (\alpha_{0:\tau} f(x_\tau) - \alpha_{0:\tau-1} f(x_{\tau-1})) - \alpha_{0:t} f(w^*) \leq \sum_{\tau=0}^{t} \alpha_\tau \nabla f(x_\tau) \cdot (w_\tau - w^*) \ .$$

Telescoping the sum in the LHS we conclude the proof,

$$\alpha_{0:t} \left( f(x_\tau) - f(w^*) \right) \leq \sum_{\tau=0}^{t} \alpha_\tau \nabla f(x_\tau) \cdot (w_\tau - w^*) \ .$$

$\square$

## E   More Intuition and Discussion Regarding the Benefit of SLowcal-SGD

**More Elaborate Intuitive Explanation.**   The intuition is the following: We have two extreme baselines: (1) Minibatch-SGD where queries do not change at all during updates-implying that there is no bias between different machines. However, Minibatch-SGD is "lazy" since among $KR$ queries it only performs $R$ gradient updates. Conversely (2) Local-SGD is not "lazy" since each machine performs $KR$ gradient updates. Nevertheless, the queries of different machines change substantially during each round, which translates to bias between machines, which in turn degrades the convergence.

Ideally, we would like to have a "non-lazy" method where each machine performs KR gradient updates (like Local-SGD), but where the queries of each machine do not change at all during rounds (like Minibatch-SGD) and therefore no bias is introduced between machines. Of course, this is too good to exist, but our method is a step in this direction: it is "non-lazy" and the query points of different machines change slowly, and therefore introduce less bias between machines. This translates to a better convergence rate.

**Additional Technical Intuition for $\alpha_t \propto t$.** Here we extend the technical explanation that we provide in Sec. 3.2.1 to the case where $\alpha_t \propto t$, and show again that SLowcal-SGD yields smaller bias between different machines compared to Local-SGD.

As in the intuition for the case of uniform weights, to simplify the more technical discussion, we will assume the homogeneous case, i.e., that for any $i \in [M]$ we have $\mathcal{D}_i = \mathcal{D}$ and $f_i(\cdot) = f(\cdot)$.

Note that upon employing linear weights, the normalization factor $\alpha_{0:T}$ that plays a major role in the convergence guarantees of Anytime-SGD (see Thm. 1) also grows as $\alpha_{0:T} \propto T^2$. Thus, in order to make an proper comparison, we should compare the bias of weighted Anytime-SGD, to the appropriate *weighted* version of SGD; where the normalization factor $\alpha_{0:T}$ also plays a similar role in the guarantees (see e.g. [35]). This weighted SGD is as follows [35], $\forall t \geq 0$

$$w_{t+1} = w_t - \eta \alpha_t g_t \; ; \quad \text{where } g_t \text{ is unbiased of } \nabla f(w_t) \,. \tag{20}$$

and after $t$ iterations it outputs $\overline{w}_T = \frac{1}{\alpha_{0:T}} \sum_{t=0}^{T} \alpha_t w_t$. And for $\alpha_t = t+1$ this version enjoys the same guarantees as standard SGD.

Next, we compare the Local-SGD version of the above weighted SGD (Eq. (20)) to our SLowcal-SGDwhen both employ $\alpha_t = t+1$. So a bit more formally, let us discuss the bias between query points in a given round $r \in [R]$, and let us denote $t_0 = rK$. The following holds for weighted **Local SGD**,

$$w_t^i = w_{t_0} - \eta \sum_{\tau=t_0}^{t-1} \alpha_\tau g_\tau^i \,, \quad \forall i \in [M], t \in [t_0, t_0 + K] \,. \tag{21}$$

where $g_\tau^i$ is the noisy gradients that Machine $i$ computes in $w_\tau^i$, and we can write $g_\tau^i := \nabla f(w_\tau^i) + \xi_\tau^i$, where $\xi_\tau^i$ is the noisy component of the gradient. Thus, for two machines $i \neq j$ we can write,

$$\mathbf{E}\|w_t^i - w_t^j\|^2 = \eta^2 \mathbf{E} \left\| \sum_{\tau=t_0}^{t-1} \alpha_\tau (g_\tau^i - g_\tau^j) \right\|^2 \approx \eta^2 \mathbf{E} \left\| \sum_{\tau=t_0}^{t-1} \alpha_\tau (\nabla f(w_\tau^i) - \nabla f(w_\tau^j)) \right\|^2 + \eta^2 \mathbf{E} \left\| \sum_{\tau=t_0}^{t-1} \alpha_\tau (\xi_\tau^i - \xi_\tau^j) \right\|^2$$

Now, we can be generous with respect to weighted SGD and only take the second noisy term into account and neglect the first term [6]. Thus we obtain,

$$\frac{1}{\eta^2} \mathbf{E}\|w_t^i - w_t^j\|^2 \lesssim \mathbf{E} \left\| \sum_{\tau=t_0}^{t-1} \alpha_\tau (\xi_\tau^i - \xi_\tau^j) \right\|^2 \approx \alpha_{t_0+K}^2 \mathbf{E} \left\| \sum_{\tau=t_0}^{t-1} (\xi_\tau^i - \xi_\tau^j) \right\|^2 \approx (rK)^2 \cdot (t - t_0) \leq r^2 K^3 \,. \tag{22}$$

where we used $\alpha_t \leq \alpha_{t_0+K}$ , $\forall t \leq t_0 + K$, as well as $\alpha_{t_0+K}^2 = (r(K+1)+1)^2 \approx (rK)^2$. We also used $t - t_0 \lessapprox K$.

Similarly, for **SLowcal-SGD** we would like to bound $\mathbf{E}\|x_t^i - x_t^j\|^2$ for two machines $i \neq j$; while assuming linear weights i.e., $\alpha_t = t+1$ , $\forall t \in [T]$. Now the update rule for the iterates $w_t^i$, is of the same form as in Eq. (21), only now $g_\tau^i := \nabla f(x_\tau^i) + \xi_\tau^i$, where $\xi_\tau^i$ is the noisy component of the gradient. Consequently, we can show the following,

$$\sum_{\tau=t_0}^{t} \alpha_\tau (w_\tau^i - w_\tau^j) \approx -\eta \sum_{\tau=t_0}^{t} \alpha_\tau \sum_{n=t_0}^{\tau-1} \alpha_n (g_n^i - g_n^j) \approx -\eta \sum_{n=t_0}^{t-1} \alpha_{n+1:t} \alpha_n (g_n^i - g_n^j) \approx -\eta r^2 K^3 \sum_{\tau=t_0}^{t-1} (g_\tau^i - g_\tau^j) \,,$$

where we took a crude approximation of $\alpha_{n+1:t} \alpha_n \approx \alpha_{t_0:t+K} \alpha_{t_0} \lessapprox rK^2 \cdot rK = r^2 K^3$. In the last "$\approx$" we also change the notation of summation variable from $n$ to $\tau$.

---

[6]It was shown in [36], that the noisy term is dominant for standard SGD

Now, by definition, $x_t^i \approx \frac{\alpha_{0:t_0}}{\alpha_{0:t}} \cdot x_{t_0} + \frac{1}{\alpha_{0:t}} \sum_{\tau=t_0}^{t} \alpha_\tau w_\tau^i$ , $\forall i \in [M], t \in [t_0, t_0 + K]$ . Thus, for two machines $i \neq j$ we have,

$$\frac{1}{\eta^2} \mathbf{E}\|x_t^i - x_t^j\|^2 = \frac{1}{\eta^2} \mathbf{E} \left\| \frac{1}{\alpha_{0:t}} \sum_{\tau=t_0}^{t} \alpha_\tau (w_\tau^i - w_\tau^j) \right\|^2 \approx \frac{1}{\eta^2} \cdot \frac{\eta^2 r^4 K^6}{(\alpha_{0:t})^2} \mathbf{E} \left\| \sum_{\tau=t_0}^{t-1} g_\tau^i - g_\tau^j \right\|^2$$

$$\approx \frac{1}{\eta^2} \cdot \frac{\eta^2 r^4 K^6}{r^4 K^4} \mathbf{E} \left\| \sum_{\tau=t_0}^{t-1} g_\tau^i - g_\tau^j \right\|^2$$

$$\approx K^2 \mathbf{E} \left\| \sum_{\tau=t_0}^{t-1} \nabla f(x_\tau^i) - \nabla f(x_\tau^j) \right\|^2 + K^2 \mathbf{E} \left\| \sum_{\tau=t_0}^{t-1} \xi_\tau^i - \xi_\tau^j \right\|^2 .$$

where we have used $\alpha_{0:t} \approx \alpha_{0:t_0} \propto t_0^2 \approx r^2 K^2$.

As we show in our analysis, the noisy term is dominant, so we can therefore bound,

$$\frac{1}{\eta^2} \mathbf{E}\|x_t^i - x_t^j\|^2 \lesssim K^2 \mathbf{E} \left\| \sum_{\tau=t_0}^{t-1} \xi_\tau^i - \xi_\tau^j \right\|^2 \approx K^2(t - t_0) \leq K^3 . \tag{23}$$

Thus Equations (22), (23), illustrate that for $\alpha_t \propto t$, then the bias of SLowcal-SGD is smaller by a factor of $r^2$ compared to the bias of weighted Local-SGD.
Since $r^2$ can be as big as $R^2$ this coincides with the benefit of SLowcal-SGD over standard SGD in the case where $\alpha_t = 1$, which we demonstrate in the main text.

Finally, note that upon dividing by the normalization factor $\alpha_{0:T}$,we have, that for SLowcal-SGD with either $\alpha_t = 1$ or $\alpha_t \propto t$ then,

$$\frac{1}{\alpha_{0:T}} \frac{1}{\eta} \mathbf{E}\|x_t^i - x_t^j\| \approx \frac{1}{R^2 K^2} \cdot \sqrt{K^3} \approx \frac{1}{RK} \cdot \sqrt{\frac{K}{R^2}} = \frac{1}{\sqrt{K} R^2} \tag{24}$$

Comparably, upon dividing by the normalization factor $\alpha_{0:T}$,we have, that for Local-SGD with either $\alpha_t = 1$ or $\alpha_t \propto t$ that,

$$\frac{1}{\alpha_{0:T}} \frac{1}{\eta} \mathbf{E}\|w_t^i - w_t^j\| \approx \frac{1}{R^2 K^2} \cdot \sqrt{R^2 K^3} \approx \frac{1}{RK} \cdot \sqrt{K} = \frac{1}{\sqrt{K} R} \tag{25}$$

Thus, with respect to the approximate and intuitive analysis that we make here SLowcal-SGD maintains similar benefit over Local-SGD for both $\alpha_t = 1$ and $\alpha_t = t + 1$.

As we explain in Appendix L, taking $\alpha_t = 1$ in SLowcal-SGD does not actually enable to provide a benefit over Local-SGD. The reason is that for $\alpha_t = 1$, the condition for which the dominant term in the bias (between different machines) is the noisy term (this enables the approximate analysis that we make here and in the body of the paper), leads to limitation on the learning rate which in turn degrades the performance for SLowcal-SGD with $\alpha_t = 1$. Conversely, for $\alpha_t \propto t$ there is no such degradation due to the limitation of the learning rate. For more details and intuition please see Appendix L.

## F  Proof of Thm. 2

*Proof of Thm. 2.* As a starting point for the analysis, for every iteration $t \in [T]$ we will define the averages of $(w_t^i, x_t^i, g_t^i)$ across all machines as follows,

$$w_t := \frac{1}{M} \sum_{i \in [M]} w_t^i , \quad \& \quad x_t := \frac{1}{M} \sum_{i \in [M]} x_t^i \quad \& \quad g_t := \frac{1}{M} \sum_{i \in [M]} g_t^i .$$

Note that Alg. 2 explicitly computes $(w_t, x_t)$ only once every $K$ local updates, and that theses are identical to the local copies of every machine at the beginning of every round. Combining the above definitions with Eq. (6) yields,

$$w_{t+1} = w_t - \eta \alpha_t g_t , \ \forall t \in [T] \tag{26}$$

Further combining these definitions with Eq. (7) yields,

$$x_{t+1} = (1 - \frac{\alpha_{t+1}}{\alpha_{0:t+1}})x_t + \frac{\alpha_{t+1}}{\alpha_{0:t+1}}w_{t+1} \, , \, \forall t \in [T] \tag{27}$$

And the above implies that the $\{x_t\}_{t\in[T]}$ sequence is an $\{\alpha_t\}_{t\in[T]}$ weighted average of $\{w_t\}_{t\in[T]}$. This enables to employ Thm. 1 which yields,

$$\alpha_{0:t}\Delta_t := \alpha_{0:t}(f(x_t) - f(w^*)) \leq \sum_{\tau=0}^{t} \alpha_\tau \nabla f(x_\tau) \cdot (w_\tau - w^*) \, ,$$

where we denote $\Delta_t := f(x_t) - f(w^*)$. The above bound highlights the challenge in the analysis: our algorithm does not directly computes unbiased estimates of $x_t$, except for the first iterate of each round. Concretely, Eq. (26) demonstrates that our algorithm effectively updates using $g_t$ which might be a biased estimate of $\nabla f(x_t)$.

It is therefore natural to decompose $\nabla f(x_\tau) = g_\tau + (\nabla f(x_\tau) - g_\tau)$ in the above bound, leading to,

$$\alpha_{0:t}\Delta_t \leq \underbrace{\sum_{\tau=0}^{t} \alpha_\tau g_\tau \cdot (w_\tau - w^*)}_{(A)} + \underbrace{\sum_{\tau=0}^{t} \alpha_\tau (\nabla f(x_\tau) - g_\tau) \cdot (w_\tau - w^*)}_{(B)} \tag{28}$$

Thus, we intend to bound the weighted error $\alpha_{0:t}\Delta_t$ by bounding two terms: (A) which is directly related to the update rule of the algorithm (Eq. (26)), and (B) which accounts for the bias between $g_t$ and $\nabla f(x_t)$.

**Notation:** In what follows we will find the following notation useful,

$$\bar{g}_t := \frac{1}{M} \sum_{i\in[M]} \nabla f_i(x_t^i) \tag{29}$$

and the above definition implies that $\bar{g}_t = \mathbf{E}\left[g_t | \{z_0^i\}_{i\in[M]}, \ldots, \{z_{t-1}^i\}_{i\in[M]}\right] = \mathbf{E}\left[g_t | \{x_t^i\}_{i\in[M]}\right]$. We will also employ the following notations,

$$V_t := \sum_{\tau=0}^{t} \alpha_\tau^2 \|\bar{g}_\tau - \nabla f(x_\tau)\|^2 \, , \quad \& \quad D_t := \|w_t - w^*\|^2$$

where $w^*$ is a global minimum of $f(\cdot)$. Moreover, we will also use the notation $D_{0:t} := \sum_{\tau=0}^{t} \|w_\tau - w^*\|^2$.

**Bounding (A):** Due to the update rule of Eq. (26), one can show by standard regret analysis (see Lemma 2 below) that,

$$(A) := \sum_{\tau=0}^{t} \alpha_\tau g_\tau \cdot (w_\tau - w^*) \leq \frac{\|w_0 - w^*\|^2}{2\eta} + \frac{\eta}{2} \sum_{\tau=0}^{t} \alpha_\tau^2 \|g_\tau\|^2 \, , \tag{30}$$

**Lemma 2.** *(OGD Regret Lemma -See e.g. [15]) Let $w_0 \in \mathbb{R}^d$ and $\eta > 0$. Also assume a sequence of $T$ non-negative weights $\{\alpha_t \geq 0\}_{t\in[T]}$ and $T$ vectors $\{g_t \in \mathbb{R}^d\}_{t\in[T]}$, and assume an update rule of the following form:*

$$w_{t+1} = w_t - \eta\alpha_t g_t \, , \forall t \in [T] \, .$$

*Then the following bound holds for any $u \in \mathbb{R}^d$, and $t \in [T]$,*

$$\sum_{\tau=0}^{t} \alpha_\tau g_\tau \cdot (w_\tau - u) \leq \frac{\|w_0 - u\|^2}{2\eta} + \frac{\eta}{2} \sum_{\tau=0}^{t} \alpha_\tau^2 \|g_\tau\|^2 \, .$$

for completeness we provide a proof in Appendix G.

**Bounding** (B): Since our goal is to bound the expected excess loss, we will bound the expected value of (B), thus,

$$
\begin{aligned}
\mathbf{E}\left[(B)\right] &= \mathbf{E}\left[\sum_{\tau=0}^{t} \alpha_\tau (\nabla f(x_\tau) - g_\tau) \cdot (w_\tau - w^*)\right] \\
&= \mathbf{E}\left[\sum_{\tau=0}^{t} \alpha_\tau (\nabla f(x_\tau) - \bar{g}_\tau) \cdot (w_\tau - w^*)\right] \\
&\leq \mathbf{E}\sum_{\tau=0}^{t} \left(\frac{1}{2\rho}\alpha_\tau^2 \|\nabla f(x_\tau) - \bar{g}_\tau\|^2 + \frac{\rho}{2}\mathbf{E}\|w_\tau - w^*\|^2\right) \\
&= \frac{1}{2\rho}\mathbf{E}V_t + \frac{\rho}{2}\mathbf{E}D_{0:t} ,
\end{aligned}
\tag{31}
$$

where the second line follows by the definition of $\bar{g}_\tau$ (see Eq. (29)) and due to the fact that $w_\tau$ is measurable with respect to $\{\{z_0^i\}_{i\in[M]}, \ldots, \{z_{\tau-1}^i\}_{i\in[M]}\}$ while $\bar{g}_\tau = \mathbf{E}\left[g_\tau | \{z_0^i\}_{i\in[M]}, \ldots, \{z_{\tau-1}^i\}_{i\in[M]}\right]$ implying that $\mathbf{E}[g_\tau \cdot (w_\tau - w^*)] = \mathbf{E}[\bar{g}_\tau \cdot (w_\tau - w^*)]$; the third line uses Young's inequality $a \cdot b \leq \inf_{\rho>0}\{\frac{\rho}{2}\|a\|^2 + \frac{1}{2\rho}\|b\|^2\}$ which holds for any $a, b \in \mathbb{R}^d$; and the last two lines use the definition of $V_t$ and $D_{0:T}$.

**Combining** (A) **and** (B): Combining Equations (30) and (31) into Eq. (28) we obtain the following bound which holds for any $\rho > 0$ and $t \in [T]$,

$$
\alpha_{0:t}\mathbf{E}\Delta_t \leq \frac{\|w_0 - w^*\|^2}{2\eta} + \frac{\eta}{2}\mathbf{E}\sum_{\tau=0}^{T} \alpha_\tau^2\|g_\tau\|^2 + \frac{1}{2\rho}\mathbf{E}V_T + \frac{\rho}{2}\mathbf{E}D_{0:t}
\tag{32}
$$

where we have used $V_t \leq V_T$ which holds for any $t \in [T]$, as well as $\mathbf{E}\sum_{\tau=0}^{t} \alpha_\tau^2\|g_\tau\|^2 \leq \mathbf{E}\sum_{\tau=0}^{T} \alpha_\tau^2\|g_\tau\|^2$, which holds since $t \leq T$.

Next, we shall bound each of the above terms. The following lemma bounds $\mathbf{E}D_{0:t}$,

**Lemma 3.** *The following bound holds for any $t \in [T]$,*

$$
\mathbf{E}D_{0:t} = \mathbf{E}\sum_{\tau=0}^{t} \|w_\tau - w^*\|^2 \leq 2T\|w_0 - w^*\|^2 + 2T\eta^2\mathbf{E}\sum_{t=0}^{T} \alpha_t^2\|g_t\|^2 + 16\eta^2 T^2 \cdot \mathbf{E}V_T
$$

Combining the above lemma into Eq. (33) gives,

$$
\alpha_{0:t}\mathbf{E}\Delta_t \leq \frac{\|w_0 - w^*\|^2}{2\eta} + \left(\frac{\eta}{2} + \rho T\eta^2\right)\mathbf{E}\sum_{\tau=0}^{T} \alpha_\tau^2\|g_\tau\|^2 + \left(\frac{1}{2\rho} + 8\rho\eta^2 T^2\right)\mathbf{E}V_T + \rho T\|w_0 - w^*\|^2
$$

Since the above holds for any $\rho > 0$ let us pick a specific value of $\rho = \frac{1}{4\eta T}$; by doing so we obtain,

$$
\alpha_{0:t}\mathbf{E}\Delta_t \leq \frac{\|w_0 - w^*\|^2}{\eta} + \eta \cdot \underbrace{\mathbf{E}\sum_{\tau=0}^{T} \alpha_\tau^2\|g_\tau\|^2 + 4\eta T\mathbf{E}V_T}_{(C)} .
\tag{33}
$$

Next, we would like to bound (C); to do so it is natural to decompose $g_\tau = (g_\tau - \bar{g}_\tau) + (\bar{g}_\tau - \nabla f(x_\tau)) + \nabla f(x_\tau)$. The next lemma provides a bound, and its proof goes directly through this decomposition,

**Lemma 4.** *The following holds,*

$$
(C) \leq 3\frac{\sigma^2}{M}\sum_{t=0}^{T} \alpha_t^2 + 3\mathbf{E}V_T + 12L\mathbf{E}\sum_{t=0}^{T} \alpha_{0:t}\Delta_t .
$$

Combining the above Lemma into Eq. (33) yields,

$$\alpha_{0:t}\mathbf{E}\Delta_t \leq \frac{\|w_0 - w^*\|^2}{\eta} + 3\eta\frac{\sigma^2}{M}\sum_{t=0}^{T}\alpha_t^2 + 8\eta T\mathbf{E}V_T + 12\eta L\mathbf{E}\sum_{t=0}^{T}\alpha_{0:t}\Delta_t \ , \tag{34}$$

where we have uses $3 \leq 4T$ which holds since $T \geq 1$. The next lemma provides a bound for $\mathbf{E}V_t$,

**Lemma 5.** *For any $t \leq T := KR$, Alg. 2 with the learning choice in Eq. (8) ensures the following bound,*

$$\mathbf{E}V_t \leq 400L^2\eta^2 K^3\sum_{\tau=0}^{T}\alpha_{0:\tau}\cdot(G_*^2 + 4L\Delta_\tau) + 90L^2\eta^2 K^6 R^3\sigma^2 \ .$$

Plugging the above bound back into Eq. (34) gives an almost explicit bound,

$\alpha_{0:t}\mathbf{E}\Delta_t$

$$\leq \frac{\|w_0 - w^*\|^2}{\eta} + 3\eta\frac{\sigma^2}{M}\sum_{t=0}^{T}\alpha_t^2 + 720L^2\eta^3 TK^6 R^3\sigma^2$$

$$+ 4\cdot 10^3 L^2\eta^3 TK^3\sum_{\tau=0}^{T}\alpha_{0:\tau}\cdot(G_*^2 + 4L\Delta_\tau)) + 12\eta L\mathbf{E}\sum_{t=0}^{T}\alpha_{0:t}\Delta_t$$

$$= \frac{\|w_0 - w^*\|^2}{\eta} + 3\eta\frac{\sigma^2}{M}\sum_{t=0}^{T}\alpha_t^2 + 720L^2\eta^3 TK^6 R^3\sigma^2 + 4\cdot 10^3 L^2\eta^3 TK^3\sum_{\tau=0}^{T}\alpha_{0:\tau}G_*^2$$

$$+ (12\eta L + 16\cdot 10^3 L^3\eta^3 TK^3)\mathbf{E}\sum_{t=0}^{T}\alpha_{0:t}\Delta_t$$

$$\leq \frac{\|w_0 - w^*\|^2}{\eta} + 3\eta\frac{\sigma^2}{M}\sum_{t=0}^{T}\alpha_t^2 + 720L^2\eta^3 TK^6 R^3\sigma^2 + 4\cdot 10^3 L^2\eta^3 TK^3\sum_{\tau=0}^{T}\alpha_{0:\tau}G_*^2 + \frac{1}{2(T+1)}\mathbf{E}\sum_{t=0}^{T}\alpha_{0:t}\Delta_t \ , \tag{35}$$

and we used $12\eta L \leq 1/4(T+1)$ which follows since $\eta \leq \frac{1}{48L(T+1)}$ (see Eq. (8)), as well as $16\cdot 10^3 L^3\eta^3 TK^3 \leq 1/4(T+1)$, which follows since $\eta \leq \frac{1}{40LK(T+1)^{2/3}}$ (see Eq. (8)). Next we use the above bound and invoke the following lemma,

**Lemma 6.** *Let $\{A_t\}_{t\in[T]}$ be a sequence of non-negative elements and $\mathcal{B} \in \mathbb{R}$, and assume that for any $t \leq T$,*

$$A_t \leq \mathcal{B} + \frac{1}{2(T+1)}\sum_{t=0}^{T}A_t \ ,$$

*Then the following bound holds,*

$$A_T \leq 2\mathcal{B} \ .$$

Taking $A_t \leftarrow \alpha_{0:t}\mathbf{E}\Delta_t$ and $\mathcal{B} \leftarrow \frac{\|w_0 - w^*\|^2}{\eta} + 3\eta\frac{\sigma^2}{M}\sum_{t=0}^{T}\alpha_t^2 + 720L^2\eta^3 TK^6 R^3\sigma^2 + 2\cdot 10^3 L^2\eta^3 TK^3\sum_{\tau=0}^{T}\alpha_{0:\tau}G_*^2$ provides the following explicit bound,

$$\alpha_{0:T}\mathbf{E}\Delta_T \leq \frac{2\|w_0 - w^*\|^2}{\eta} + 6\eta\frac{\sigma^2}{M}\sum_{t=0}^{T}\alpha_t^2 + 2\cdot 10^3 L^2\eta^3 TK^6 R^3\sigma^2 + 8\cdot 10^3 L^2\eta^3 TK^3\sum_{\tau=0}^{T}\alpha_{0:\tau}G_*^2$$

$$\leq \frac{2\|w_0 - w^*\|^2}{\eta} + 6\eta\frac{\sigma^2}{M}\cdot(KR)^3 + 2\cdot 10^3 L^2\eta^3 K^7 R^4\sigma^2 + 8\cdot 10^3 L^2\eta^3 K^7 R^4\cdot G_*^2 \ , \tag{36}$$

where we have used $\sum_{\tau=0}^{T}\alpha_{0:\tau} \leq \sum_{t=0}^{T}\alpha_t^2 \leq \sum_{r=0}^{R-1}\sum_{k=0}^{K-1}(r+1)^2 K^2 \leq K^3 R^3$, as well as $T = KR$,

Recalling that $T = KR$ and that,

$$\eta = \min\left\{\frac{1}{48L(T+1)}, \frac{1}{10LK^2}, \frac{1}{40LK(T+1)^{2/3}}, \frac{\|w_0 - w^*\|\sqrt{M}}{\sigma T^{3/2}}, \frac{\|w_0 - w^*\|^{1/2}}{L^{1/2}K^{7/4}R(\sigma^{1/2} + G_*^{1/2})}\right\}$$

The above bound translates into,

$$\alpha_{0:T}\mathbf{E}\Delta_T \leq \tag{37}$$

$$O\left(L(T + K^2 + KT^{2/3})\|w_0 - w^*\|^2 + \frac{\sigma\|w_0 - w^*\|T^{3/2}}{\sqrt{M}} + L^{1/2}K^{7/4}R(\sigma^{1/2} + G_*^{1/2}) \cdot \|w_0 - w^*\|^{3/2}\right) \tag{38}$$

Noting that $\alpha_{0:T} \geq \Omega(T^2)$ and using $T = KR$ gives the final bound,

$$\mathbf{E}\Delta_T \leq$$

$$O\left(\frac{L\|w_0 - w^*\|^2}{KR} + \frac{L\|w_0 - w^*\|^2}{K^{1/3}R^{4/3}} + \frac{L\|w_0 - w^*\|^2}{R^2} + \frac{\sigma\|w_0 - w^*\|}{\sqrt{MKR}} + \frac{L^{1/2}(\sigma^{1/2} + G_*^{1/2}) \cdot \|w_0 - w^*\|^{3/2}}{K^{1/4}R}\right).$$

which establishes the Theorem. $\qquad\square$

## G  Proof of Lemma 2

*Proof of Lemma 2.* The update rule implies for all $\tau \in [T]$

$$\|w_{\tau+1} - u\|^2 = \|(w_\tau - u) - \eta\alpha_\tau g_\tau\|^2$$
$$= \|w_\tau - u\|^2 - 2\eta\alpha_\tau g_\tau \cdot (w_\tau - u) + \eta^2\alpha_\tau^2\|g_\tau\|^2$$

Re-ordering and gives,

$$2\eta\alpha_\tau g_\tau \cdot (w_\tau - u) = \left(\|w_\tau - u\|^2 - \|w_{\tau+1} - u\|^2\right) + \eta^2\alpha_\tau^2\|g_\tau\|^2.$$

Summing over $\tau$ and telescoping we obtain,

$$2\eta\sum_{\tau=0}^{t}\alpha_\tau g_\tau \cdot (w_\tau - u) = \left(\|w_1 - u\|^2 - \|w_{t+1} - u\|^2\right) + \eta^2\sum_{\tau=0}^{t}\alpha_\tau^2\|g_\tau\|^2$$

$$\leq \|w_1 - u\|^2 + \eta^2\sum_{\tau=0}^{t}\alpha_\tau^2\|g_\tau\|^2$$

Dividing the above by $2\eta$ establishes the lemma. $\qquad\square$

## H  Proof of Lemma 3

*Proof of Lemma 3.* Recalling the notations $D_\tau := \|w_\tau - w^*\|^2$, our goal is to bound $\mathbf{E}D_{0:t}$. To do so, we will derive a recursive formula for $D_{0:t}$. Indeed, the update rule of Alg. 2 implies Eq. (26), which in turn leads to the following for any $t \in [T]$,

$$\|w_{t+1} - w^*\|^2 = \|(w_t - w^*) - \eta\alpha_t g_t\|^2 = \|w_t - w^*\|^2 - 2\eta\alpha_t g_t \cdot (w_t - w^*) + \eta^2\alpha_t^2\|g_t\|^2$$

Unrolling the above equation and taking expectation gives,

$$\mathbf{E}\|w_{t+1} - w^*\|^2 = \|w_0 - w^*\|^2 \underbrace{- 2\eta\mathbf{E}\sum_{\tau=0}^{t}\alpha_\tau g_\tau \cdot (w_\tau - w^*)}_{(*)} + \eta^2\mathbf{E}\sum_{\tau=0}^{t}\alpha_\tau^2\|g_\tau\|^2 \tag{39}$$

The next lemma provides a bound on $(*)$,

**Lemma 7.** *The following holds for any $t \in [T]$,*

$$(*) \leq 2\eta\sqrt{\mathbf{E}V_T} \cdot \sqrt{\mathbf{E}D_{0:T}}$$

*and recall that $D_{0:T} := \sum_{t=0}^{T}\|w_t - w^*\|^2$, and $V_T := \sum_{t=0}^{T}\alpha_t^2\|\bar{g}_t - \nabla f(x_t)\|^2$.*

Plugging the bound of Lemma 7 into Eq. (39), and using the notation of $D_t$ we conclude that for any $t \in [T]$,

$$\mathbf{E}D_t \le D_0 + 2\eta\sqrt{\mathbf{E}V_T} \cdot \sqrt{\mathbf{E}D_{0:T}} + \eta^2\mathbf{E}\sum_{\tau=0}^{t}\alpha_\tau^2\|g_\tau\|^2$$

$$\le D_0 + 2\eta\sqrt{\mathbf{E}V_T} \cdot \sqrt{\mathbf{E}D_{0:T}} + \eta^2\mathbf{E}\sum_{t=0}^{T}\alpha_t^2\|g_t\|^2 \,,$$

where we used $t \le T$. Summing the above equation over $t$ gives,

$$\mathbf{E}D_{0:T} \le T\|w_0 - w^*\|^2 + 2\eta T \cdot \sqrt{\mathbf{E}V_T} \cdot \sqrt{\mathbf{E}D_{0:T}} + T\eta^2\mathbf{E}\sum_{t=0}^{T}\alpha_t^2\|g_t\|^2 \tag{40}$$

We shall now require the following lemma,

**Lemma 8.** *Let $A, B, C \ge 0$, and assume that $A \le B + C\sqrt{A}$, then the following holds,*
$$A \le 2B + 4C^2$$

Now, using the above Lemma with Eq. (40) implies,

$$\mathbf{E}D_{0:T} \le 2T\|w_0 - w^*\|^2 + 2T\eta^2\mathbf{E}\sum_{t=0}^{T}\alpha_t^2\|g_t\|^2 + 16\eta^2T^2 \cdot \mathbf{E}V_T \tag{41}$$

where we have taken $A \leftarrow D_{0:T}^2$, $B \leftarrow T\|w_0 - w^*\|^2 + T\eta^2\mathbf{E}\sum_{t=0}^{T}\alpha_t^2\|g_t\|^2$, and $C \leftarrow 2\eta T \cdot \sqrt{\mathbf{E}V_T}$. Thus, Eq. (41) establishes the lemma. □

## H.1 Proof of Lemma 7

*Proof of Lemma 7.* Recall that $(*) = -2\eta\mathbf{E}\sum_{\tau=0}^{t}\alpha_\tau g_\tau \cdot (w_\tau - w^*)$, we shall now focus on bounding $(*)/2\eta$,

$$-\mathbf{E}\sum_{\tau=0}^{t}\alpha_\tau g_\tau \cdot (w_\tau - w^*) = -\mathbf{E}\sum_{\tau=0}^{t}\alpha_\tau\bar{g}_\tau \cdot (w_\tau - w^*)$$

$$= -\mathbf{E}\sum_{\tau=0}^{t}\alpha_\tau\nabla f(x_\tau) \cdot (w_\tau - w^*) - \mathbf{E}\sum_{\tau=0}^{t}\alpha_\tau(\bar{g}_\tau - \nabla f(x_\tau)) \cdot (w_\tau - w^*)$$

$$\le 0 + \mathbf{E}\sum_{\tau=0}^{t}\alpha_\tau^2\|\bar{g}_\tau - \nabla f(x_\tau)\|^2 \cdot \|w_\tau - w^*\|^2$$

$$\le 0 + \sum_{\tau=0}^{t}\sqrt{\mathbf{E}\alpha_\tau^2\|\bar{g}_\tau - \nabla f(x_\tau)\|^2} \cdot \sqrt{\mathbf{E}\|w_\tau - w^*\|^2}$$

$$\le \sqrt{\mathbf{E}\sum_{\tau=0}^{t}\alpha_\tau^2\|\bar{g}_\tau - \nabla f(x_\tau)\|^2} \cdot \sqrt{\mathbf{E}\sum_{\tau=0}^{t}\|w_\tau - w^*\|^2}$$

$$\le \sqrt{\mathbf{E}\sum_{t=0}^{T}\alpha_t^2\|\bar{g}_t - \nabla f(x_t)\|^2} \cdot \sqrt{\mathbf{E}\sum_{t=0}^{T}\|w_t - w^*\|^2}$$

$$:= \sqrt{\mathbf{E}V_T} \cdot \sqrt{\mathbf{E}D_{0:T}} \,, \tag{42}$$

where the first line is due to the definitions of $g_\tau$ and $\bar{g}_\tau$ appearing in Eq. (29) (this is formalized in Lemma 9 below and in its proof); the third line follows by observing that the $\{x_t\}_t$ sequence is and $\{\alpha_t\}_t$ weighted average of $\{w_t\}_t$ and thus Theorem 1 implies that $\mathbf{E}\sum_{\tau=0}^{t}\alpha_\tau\nabla f(x_\tau) \cdot (w_\tau -$

$w^*) \geq 0$ for any $t$, as well as from Cauchy-Schwarz; the fourth line follows from the Cauchy-Schwarz inequality for random variables, which asserts that for every random variables $X, Y$, then $\mathbf{E}[XY] \leq \sqrt{\mathbf{E}X^2}\sqrt{\mathbf{E}Y^2}$; the fifth line is an application of the following inequality

$$\sum_{\tau=0}^{t} a_\tau b_\tau \leq \sqrt{\sum_{\tau=0}^{t} a_\tau^2}\sqrt{\sum_{\tau=0}^{t} b_\tau^2}$$

which holds for any two sequences $\{a_\tau \in \mathbb{R}\}_\tau, \{b_\tau \in \mathbb{R}\}_\tau$, and the above also follows from the standard Cauchy-Schwarz inequality. Thus, Eq. (42) establishes the lemma.

We are left to show that $\mathbf{E}[g_\tau \cdot (w_\tau - w^*)] = \mathbf{E}[\bar{g}_\tau \cdot (w_\tau - w^*)]$ which is established in the lemma below,

**Lemma 9.** *The following holds for any* $\tau \in [T]$,

$$\mathbf{E}[g_\tau \cdot (w_\tau - w^*)] = \mathbf{E}[\bar{g}_\tau \cdot (w_\tau - w^*)] \ .$$

□

### H.1.1 Proof of Lemma 9

*Proof of Lemma 9.* Let $\{\mathcal{F}_\tau\}_{\tau \in [T]}$ be the natural filtration induces by the history of samples up to every time step $\tau$. Then according to the definitions of $g_t$ and $\bar{g}_t$ we have,

$$\begin{aligned}
\mathbf{E}[g_\tau \cdot (w_\tau - w^*)] &= \mathbf{E}[\mathbf{E}[g_\tau \cdot (w_\tau - w^*)|\mathcal{F}_{\tau-1}]] \\
&= \mathbf{E}[\mathbf{E}[g_\tau|\mathcal{F}_{\tau-1}] \cdot (w_\tau - w^*)] \\
&= \mathbf{E}[\mathbf{E}[\bar{g}_\tau|\mathcal{F}_{\tau-1}] \cdot (w_\tau - w^*)] \\
&= \mathbf{E}[\bar{g}_\tau \cdot (w_\tau - w^*)] \ ,
\end{aligned}$$

where the first line follows by the law of total expectations; the second line follows since $w_\tau$ is measurable w.r.t. $\mathcal{F}_{\tau-1}$; the third line follows by definition of $g_\tau$ and $\bar{g}_\tau$; and the last line uses the law of total expectations. □

### H.2 Proof of Lemma 8

*Proof of Lemma 8.* We will divide the proof into two case.
**Case 1:** $B \geq C\sqrt{A}$**.** In this case,

$$A \leq B + C\sqrt{A} \leq 2B \leq 2B + 4C^2 \ .$$

**Case 2:** $B \leq C\sqrt{A}$**.** In this case,

$$A \leq B + C\sqrt{A} \leq 2C\sqrt{A} \ ,$$

dividing by $\sqrt{A}$ and taking the square implies,

$$A \leq 4C^2 \leq 2B + 4C^2 \ .$$

And therefore the lemma holds. □

## I Proof of Lemma 4

*Proof of Lemma 4.* Recalling that $(C) := \mathbf{E}\sum_{\tau=0}^{T} \alpha_\tau^2 \|g_\tau\|^2$, we will decompose $g_\tau = (g_\tau - \bar{g}_\tau) + (\bar{g}_\tau - \nabla f(x_\tau)) + \nabla f(x_\tau)$ which gives,

$$(C) := \mathbf{E} \sum_{\tau=0}^{T} \alpha_\tau^2 \| (g_\tau - \bar{g}_\tau) + (\bar{g}_\tau - \nabla f(x_\tau)) + \nabla f(x_\tau) \|^2$$

$$\leq 3\mathbf{E} \sum_{\tau=0}^{T} \alpha_\tau^2 \| g_\tau - \bar{g}_\tau \|^2 + 3\mathbf{E} \sum_{\tau=0}^{T} \alpha_\tau^2 \| \bar{g}_\tau - \nabla f(x_\tau) \|^2 + 3\mathbf{E} \sum_{\tau=0}^{T} \alpha_\tau^2 \| \nabla f(x_\tau) \|^2$$

$$\leq 3\mathbf{E} \sum_{\tau=0}^{T} \alpha_\tau^2 \| g_\tau - \bar{g}_\tau \|^2 + 3\mathbf{E} V_T + 6L\mathbf{E} \sum_{\tau=0}^{T} \alpha_\tau^2 \Delta_\tau$$

$$\leq 3\frac{\sigma^2}{M} \mathbf{E} \sum_{\tau=0}^{T} \alpha_\tau^2 + 3\mathbf{E} V_T + 6L\mathbf{E} \sum_{\tau=0}^{T} \alpha_\tau^2 \Delta_\tau$$

$$\leq 3\frac{\sigma^2}{M} \mathbf{E} \sum_{\tau=0}^{T} \alpha_\tau^2 + 3\mathbf{E} V_T + 12L\mathbf{E} \sum_{\tau=0}^{T} \alpha_{0:\tau} \Delta_\tau \ , \tag{43}$$

where the second line uses $\|a + b + c\|^2 \leq 3(\|a\|^2 + \|b\|^2 + \|c\|^2)$ which holds for any $a, b, c \in \mathbb{R}^d$; the third line uses the definition of $V_T$ as well as the smoothness of $f(\cdot)$ implying that $\|\nabla f(x_\tau)\|^2 \leq 2L(f(x_\tau) - f(w^*)) := 2L\Delta_\tau$ (see Lemma 10 below); the fourth line invokes Lemma 11; the fifth line uses $\alpha_\tau^2 = (\tau + 1)^2 \leq 2\alpha_{0:\tau}$.

**Lemma 10.** *Let $F : \mathbb{R}^d \mapsto \mathbb{R}$ be an L-smooth function with a global minimum $x^*$, then for any $x \in \mathbb{R}^d$ we have,*

$$\|\nabla F(x)\|^2 \leq 2L(F(x) - F(w^*)) \ .$$

**Lemma 11.** *The following bound holds for any $t \in [T]$,*

$$\mathbf{E}\|g_\tau - \bar{g}_\tau\|^2 \leq \frac{\sigma^2}{M} \ .$$

$\square$

## I.1 Proof of Lemma 10

*Proof of Lemma 10.* The $L$ smoothness of $f$ means the following to hold $\forall w, u \in \mathbb{R}^d$,

$$F(x + u) \leq F(x) + \nabla F(x)^\top u + \frac{L}{2} \|u\|^2 \ .$$

Taking $u = -\frac{1}{L} \nabla F(x)$ we get,

$$F(x + u) \ \leq \ F(x) - \frac{1}{L} \|\nabla F(x)\|^2 + \frac{1}{2L} \|\nabla F(x)\|^2 = F(x) - \frac{1}{2L} \|\nabla F(x)\|^2 \ .$$

Thus:

$$\begin{aligned} \|\nabla F(x)\|^2 \ &\leq \ 2L\big(F(x) - F(x + u)\big) \\ &\leq \ 2L\big(F(x) - F(x^*)\big) \ , \end{aligned}$$

where in the last inequality we used $F(x^*) \leq F(x + u)$ which holds since $x^*$ is the *global* minimum.

$\square$

## I.2 Proof of Lemma 11

*Proof of Lemma 11.* Recall that we can write,

$$g_\tau - \bar{g}_\tau := \frac{1}{M} \sum_{i \in [M]} (g_\tau^i - \bar{g}_\tau^i)$$

where $\bar{g}_\tau^i := \nabla f_i(x_\tau^i)$, and $g_\tau^i := \nabla f_i(x_\tau^i, z_\tau^i)$, and that $z_t^1, \ldots, z_t^M$ are independent of each other. Thus, conditioning over $\{x_t^i\}_{i=1}^M$ then $\{g_\tau^i - \bar{g}_\tau^i\}_{i=1}^M$ are independent and zero mean i.e. $\mathbf{E}[g_\tau^i - \bar{g}_\tau^i | \{x_t^i\}_{i=1}^M] = 0$. Consequently,

$$
\begin{aligned}
\mathbf{E}\left[\|g_\tau - \bar{g}_\tau\|^2 | \{x_t^i\}_{i=1}^M\right] &= \frac{1}{M^2} \mathbf{E}\left[\left\|\sum_{i \in [M]} (g_\tau^i - \bar{g}_\tau^i)\right\|^2 | \{x_t^i\}_{i \in [M]}\right] \\
&= \frac{1}{M^2} \sum_{i \in [M]} \mathbf{E}\left[\|g_\tau^i - \bar{g}_\tau^i\|^2 | \{x_t^i\}_{i=1}^M\right] \\
&\leq \frac{1}{M^2} \sum_{i \in [M]} \sigma^2 \\
&\leq \frac{\sigma^2}{M} .
\end{aligned}
$$

Using the law of total expectation implies that $\mathbf{E}\|g_\tau - \bar{g}_\tau\|^2 \leq \frac{\sigma^2}{M}$. $\qquad\square$

## J  Proof of Lemma 5

*Proof of Lemma 5.* To bound $\mathbf{E}V_t$ we will first employ the definition of $x_t$ together with the smoothness of $f(\cdot)$,

$$
\begin{aligned}
\mathbf{E}\|\nabla f(x_\tau) - \bar{g}_\tau\|^2 = \mathbf{E}\left\|\frac{1}{M}\sum_{i \in [M]} \nabla f_i(x_\tau) - \frac{1}{M}\sum_{i \in [M]} \nabla f_i(x_\tau^i)\right\|^2 \\
\leq \frac{1}{M}\sum_{i \in [M]} \mathbf{E}\|\nabla f_i(x_\tau) - \nabla f_i(x_\tau^i)\|^2 \\
\leq \frac{L^2}{M}\sum_{i \in [M]} \mathbf{E}\|x_\tau - x_\tau^i\|^2 \\
= \frac{L^2}{M}\sum_{i \in [M]} \mathbf{E}\left\|\frac{1}{M}\sum_{j \in [M]} x_\tau^j - x_\tau^i\right\|^2 \\
\leq \frac{L^2}{M^2}\sum_{i,j \in [M]} \mathbf{E}\|x_\tau^j - x_\tau^i\|^2 , \qquad (44)
\end{aligned}
$$

where the first line uses the definition of $\bar{g}_t$, the second line uses Jensen's inequality, and the third line uses the smoothness of $f_i(\cdot)$'s. The last line follows from Jensen's inequality.

We use the following notation for any $\tau \in [T]$

$$
q_\tau^{i,j} := \alpha_\tau^2 \|x_\tau^i - x_\tau^j\|^2 , \quad \& \quad q_\tau^i := \alpha_\tau^2 \sum_{j \in [M]} \|x_\tau^i - x_\tau^j\|^2 , \quad \& \quad Q_\tau := \frac{1}{M^2}\alpha_\tau^2 \sum_{i,j \in [M]} \|x_\tau^i - x_\tau^j\|^2 ,
$$

(45)

and notice that $\sum_{j \in [M]} q_\tau^{i,j} = q_\tau^i$, and that $\sum_{i,j \in [M]} q_\tau^{i,j} = M^2 Q_\tau$. Moreover $q_\tau^{i,j} = q_\tau^{j,i}, \forall i, j \in [M]$.

Thus, according to Eq. (44) it is enough to bound $\mathbf{E}V_t$ as follows,

$$
\mathbf{E}V_t := \sum_{\tau=0}^t \alpha_\tau^2 \mathbf{E}\|\nabla f(x_\tau) - \bar{g}_\tau\|^2 \leq L^2 \cdot \underbrace{\sum_{\tau=0}^t Q_\tau}_{(\star)} .
$$

(46)

Next we will bound the above term.

**Bounding ($\star$):** Let $t \in [T]$, if $t = rK$ for some $r \in [R]$, then according to Alg. 2 $x_t^i = x_t$ for any machine $i \in [M]$, thus $x_t^i - x_t^j = 0$ for any two machines $i, j \in [M]$.

More generally, if $t = rK + k$ for some $r \in [R]$, and $k \in [K]$, then by denoting $t_0 := rK$ we can write $t = t_0 + k$. Using this notation, the update rule for $x_\tau^i$ implies the following for any $i \in [M]$,

$$x_t^i = \frac{\alpha_{0:t_0}}{\alpha_{0:t}} x_{t_0}^i + \frac{1}{\alpha_{0:t}} \sum_{\tau=t_0+1}^{t} \alpha_\tau w_\tau^i = \frac{\alpha_{0:t_0}}{\alpha_{0:t}} x_{t_0} + \frac{1}{\alpha_{0:t}} \sum_{\tau=t_0+1}^{t} \alpha_\tau w_\tau^i \,,$$

where we used $x_{t_0}^i = x_{t_0}$, $\forall i \in [M]$. Thus, for any $i \neq j$ we can write,

$$\alpha_t^2 \|x_t^i - x_t^j\|^2 = \frac{\alpha_t^2}{(\alpha_{0:t})^2} \left\| \sum_{\tau=t_0+1}^{t} \alpha_\tau (w_\tau^i - w_\tau^j) \right\|^2 . \tag{47}$$

So our next goal is to derive an expression for $\sum_{\tau=t_0+1}^{t} \alpha_\tau (w_\tau^i - w_\tau^j)$. The update rule of Eq. (6) implies that for any $\tau \in [t_0, t_0 + K]$,

$$w_\tau^i = w_{t_0}^i - \eta \sum_{n=t_0+1}^{\tau} \alpha_n g_n^i = w_{t_0} - \eta \sum_{n=t_0+1}^{\tau} \alpha_n g_n^i \tag{48}$$

where the second equality is due to the initialization of each round implying that $w_{t_0}^i = w_{t_0}$, $\forall i \in [M]$.

Next, we will require the following notation $\bar{g}_t^i := \nabla f_i(x_t^i)$, and $\xi_t^i := g_t^i - \bar{g}_t^i$. We can therefore write, $g_t^i = \bar{g}_t^i + \xi_t^i$ and it is immediate to show that $\mathbf{E}[\xi_t^i | x_t^i] = 0$. Using this notation together with Eq. (48), implies that for any $\tau \in [t_0, t_0 + K]$ and $i \neq j$ we have,

$$\alpha_\tau (w_\tau^i - w_\tau^j) = -\eta \sum_{n=t_0+1}^{\tau} \alpha_\tau \alpha_n (\bar{g}_n^i - \bar{g}_n^j) - \eta \sum_{n=t_0+1}^{\tau} \alpha_\tau \alpha_n (\xi_n^i - \xi_n^j)$$

$$= -\eta \sum_{n=t_0+1}^{\tau} \alpha_\tau \alpha_n (\bar{g}_n^i - \bar{g}_n^j) - \eta \sum_{n=t_0+1}^{\tau} \alpha_\tau \alpha_n \xi_n \tag{49}$$

and in the last line we use the following notation $\xi_n := \xi_n^i - \xi_n^j$ [7].

Summing Eq. (49) over $\tau \in [t_0 + 1, t]$ we obtain,

$$\sum_{\tau=t_0+1}^{t} \alpha_\tau (w_\tau^i - w_\tau^j) = -\eta \sum_{\tau=t_0+1}^{t} \sum_{n=t_0+1}^{\tau} \alpha_\tau \alpha_n (\bar{g}_n^i - \bar{g}_n^j) - \eta \sum_{\tau=t_0+1}^{t} \sum_{n=t_0+1}^{\tau} \alpha_\tau \alpha_n \xi_n$$

$$= -\eta \sum_{n=t_0+1}^{t} \sum_{\tau=n}^{t} \alpha_\tau \alpha_n (\bar{g}_n^i - \bar{g}_n^j) - \eta \sum_{n=t_0+1}^{t} \sum_{\tau=n}^{t} \alpha_\tau \alpha_n \xi_n$$

$$= -\eta \sum_{n=t_0+1}^{t} \alpha_{n:t} \alpha_n (\bar{g}_n^i - \bar{g}_n^j) - \eta \sum_{n=t_0+1}^{t} \alpha_{n:t} \alpha_n \xi_n$$

$$= -\eta \sum_{\tau=t_0+1}^{t} \alpha_{\tau:t} \alpha_\tau (\bar{g}_\tau^i - \bar{g}_\tau^j) - \eta \sum_{\tau=t_0+1}^{t} \alpha_{\tau:t} \alpha_\tau \xi_\tau \,, \tag{50}$$

where in the last equation we replace the notation of the summation index from $n$ to $\tau$ (only done to ease notation).

---

[7] A more appropriate notation would be $\xi_n^{(i,j)} := \xi_n^i - \xi_n^j$, but to ease notation we absorb the $(i,j)$ notation into $\xi$.

Plugging the above equation back into Eq. (47) we obtain for any $t \in [t_0, t_0 + K]$

$$q_t^{i,j} := \alpha_t^2 \|x_t^i - x_t^j\|^2$$

$$= \frac{\alpha_t^2}{(\alpha_{0:t})^2} \left\| \sum_{\tau=t_0+1}^{t} \alpha_\tau (w_\tau^i - w_\tau^j) \right\|^2$$

$$= \frac{\alpha_t^2}{(\alpha_{0:t})^2} \left\| \eta \sum_{\tau=t_0+1}^{t} \alpha_{\tau:t} \alpha_\tau (\bar{g}_\tau^i - \bar{g}_\tau^j) + \eta \sum_{\tau=t_0+1}^{t} \alpha_{\tau:t} \alpha_\tau \xi_\tau \right\|^2$$

$$\leq \eta^2 \frac{2\alpha_t^2}{(\alpha_{0:t})^2} \left\| \sum_{\tau=t_0+1}^{t} \alpha_{\tau:t} \alpha_\tau (\bar{g}_\tau^i - \bar{g}_\tau^j) \right\|^2 + \eta^2 \frac{2\alpha_t^2}{(\alpha_{0:t})^2} \left\| \sum_{\tau=t_0+1}^{t} \alpha_{\tau:t} \alpha_\tau \xi_\tau \right\|^2$$

$$\leq \eta^2 \frac{2\alpha_t^2}{(\alpha_{0:t})^2} \cdot (t - t_0) \sum_{\tau=t_0+1}^{t} (\alpha_{\tau:t} \alpha_\tau)^2 \|\bar{g}_\tau^i - \bar{g}_\tau^j\|^2 + \eta^2 \frac{2\alpha_t^2}{(\alpha_{0:t})^2} \left\| \sum_{\tau=t_0+1}^{t} \alpha_{\tau:t} \alpha_\tau \xi_\tau \right\|^2$$

$$\leq \eta^2 \frac{2\alpha_t^2}{(\alpha_{0:t})^2} \cdot K \cdot (K\alpha_t)^2 \sum_{\tau=t_0+1}^{t} \alpha_\tau^2 \|\nabla f_i(x_\tau^i) - \nabla f_j(x_\tau^j)\|^2 + \eta^2 \frac{2\alpha_t^2}{(\alpha_{0:t})^2} \left\| \sum_{\tau=t_0+1}^{t} \alpha_{\tau:t} \alpha_\tau \xi_\tau \right\|^2$$

$$= \eta^2 \frac{2\alpha_t^4}{(\alpha_{0:t})^2} \cdot K^3 L^2 \cdot \frac{1}{L^2} \sum_{\tau=t_0+1}^{t} \alpha_\tau^2 \|\nabla f_i(x_\tau^i) - \nabla f_j(x_\tau^j)\|^2 + \eta^2 \frac{2\alpha_t^2}{(\alpha_{0:t})^2} \left\| \sum_{\tau=t_0+1}^{t} \alpha_{\tau:t} \alpha_\tau \xi_\tau \right\|^2$$

$$\leq 8\eta^2 K^3 L^2 \cdot \frac{1}{L^2} \sum_{\tau=t_0+1}^{t} \alpha_\tau^2 \|\nabla f_i(x_\tau^i) - \nabla f_j(x_\tau^j)\|^2 + (8\eta^2/\alpha_t^2) \left\| \sum_{\tau=t_0+1}^{t} \alpha_{\tau:t} \alpha_\tau \xi_\tau \right\|^2$$

$$= 8\eta^2 K^3 L^2 \cdot \frac{1}{L^2} \sum_{\tau=t_0+1}^{t} \alpha_\tau^2 \|\nabla f_i(x_\tau^i) - \nabla f_j(x_\tau^j)\|^2 + 8\eta^2 \alpha_t^2 \left\| \sum_{\tau=t_0+1}^{t} \alpha_{\tau:t} \frac{\alpha_\tau}{\alpha_t^2} \xi_\tau \right\|^2 ,$$
(51)

where the first inequality uses $\|a+b\|^2 \leq 2\|a\|^2 + 2\|b\|^2$ which holds $\forall a, b \in \mathbb{R}^d$, the second inequality uses $\|\sum_{n=N_1+1}^{N_2} a_n\|^2 \leq (N_2 - N_1) \sum_{n=N_1+1}^{N_2} \|a_n\|^2$ which holds for any $\{a_n \in \mathbb{R}^d\}_{n=N_1+1}^{N_2}$; the third inequality uses the definition of $\bar{g}_\tau^i$, it also uses $t - t_0 \leq K$ as well as $(\alpha_{\tau:t})^2 \leq (K\alpha_t)^2$ which holds since $\tau \leq t$ and since both $\alpha_\tau \leq \alpha_t$; and the last inequality uses the fact that $\alpha_t = t + 1$ implying that the following holds,

$$\frac{\alpha_t^4}{(\alpha_{0:t})^2} \leq 4 , \quad \& \quad \frac{\alpha_t^2}{(\alpha_{0:t})^2} \leq \frac{4}{\alpha_t^2}$$

**Lemma 12.** *The following holds for any $i, j \in [M]$,*

$$\|\nabla f_i(x_\tau^i) - \nabla f_j(x_\tau^j)\|^2 \leq \frac{3L^2}{M\alpha_\tau^2}(q_\tau^i + q_\tau^j) + 6 \left( \|\nabla f_i(x_\tau)\|^2 + \|\nabla f_j(x_\tau)\|^2 \right) ,$$

Using the above lemma inside Eq. (51) yields,

$$q_t^{i,j} := \alpha_t^2 \|x_t^i - x_t^j\|^2$$

$$\leq 24\eta^2 K^3 L^2 \cdot \frac{1}{M}(q_{t_0+1:t}^i + q_{t_0+1:t}^j) + 48\eta^2 K^3 \sum_{\tau=t_0+1}^{t} \alpha_\tau^2 \left( \|\nabla f_i(x_\tau)\|^2 + \|\nabla f_j(x_\tau)\|^2 \right)$$

$$+ 8\eta^2 \alpha_t^2 \left\| \sum_{\tau=t_0+1}^{t} \alpha_{\tau:t} \frac{\alpha_\tau}{\alpha_t^2} \xi_\tau \right\|^2$$

$$= \frac{\theta}{2} \cdot \frac{1}{M}(q_{t_0+1:t}^i + q_{t_0+1:t}^j) + \frac{\theta}{L^2} \sum_{\tau=t_0+1}^{t} \alpha_\tau^2 \left( \|\nabla f_i(x_\tau)\|^2 + \|\nabla f_j(x_\tau)\|^2 \right) + \mathcal{B}_t^{i,j} , \quad (52)$$

where we have denoted [8]

$$\theta := 48\eta^2 K^3 L^2 \;, \quad \& \quad \mathcal{B}_t^{i,j} := 8\eta^2 \alpha_t^2 \left\| \sum_{\tau=t_0+1}^{t} \alpha_{\tau:t} \frac{\alpha_\tau}{\alpha_t^2} \xi_\tau \right\|^2 .$$

Summing Eq. (52) over $i, j \in [M]$ and using the definition of $Q_t$ gives,

$$
\begin{aligned}
M^2 Q_t &= \sum_{i,j \in [M]} q_t^{i,j} \\
&\leq \frac{\theta}{2} \cdot \frac{1}{M} \cdot 2M^3 Q_{t_0+1:t} + \frac{\theta}{L^2} \cdot 2M \sum_{\tau=t_0+1}^{t} \alpha_\tau^2 \sum_{i \in [M]} \|\nabla f_i(x_\tau)\|^2 + \sum_{i,j \in [M]} \mathcal{B}_t^{i,j} \\
&= M^2 \theta \cdot Q_{t_0+1:t} + \frac{2M\theta}{L^2} \sum_{\tau=t_0+1}^{t} \alpha_\tau^2 \sum_{i \in [M]} \|\nabla f_i(x_\tau)\|^2 + \sum_{i,j \in [M]} \mathcal{B}_t^{i,j} \;,
\end{aligned}
\tag{53}
$$

where we used,

$$\sum_{i,j \in [M]} q_\tau^i = \sum_{j \in [M]} \sum_{i \in [M]} q_\tau^i = \sum_{j \in [M]} M^2 Q_\tau = M^3 Q_\tau \;.$$

Now dividing Eq. (53) by $M^2$ gives $\forall t \in [t_0, t_0 + K]$,

$$
\begin{aligned}
Q_t &\leq \theta \cdot Q_{t_0+1:t} + \frac{2\theta}{L^2} \sum_{\tau=t_0+1}^{t} \alpha_\tau^2 \cdot \frac{1}{M} \sum_{i \in [M]} \|\nabla f_i(x_\tau)\|^2 + \frac{1}{M^2} \sum_{i,j \in [M]} \mathcal{B}_t^{i,j} \\
&\leq \theta \cdot Q_{t_0+1:t} + \frac{2\theta}{L^2} \sum_{\tau=t_0+1}^{t} \alpha_\tau^2 \cdot (G_*^2 + 4L(f(x_\tau) - f(w^*))) + \frac{1}{M^2} \sum_{i,j \in [M]} \mathcal{B}_t^{i,j} \\
&\leq \theta \cdot Q_{t_0+1:t} + \frac{2\theta}{L^2} \sum_{\tau=t_0+1}^{t} \alpha_\tau^2 \cdot (G_*^2 + 4L\Delta_\tau) + \frac{1}{M^2} \sum_{i,j \in [M]} \mathcal{B}_t^{i,j} \;,
\end{aligned}
\tag{54}
$$

where the second line follows from the dissimilarity assumption Eq. (2), and the last line is due to the definition of $\Delta_\tau$.

Thus, we can re-write the above equation as follows forall $t \in [t_0, t_0 + K]$,

$$Q_t \leq \theta Q_{t_0+1:t} + H_t \;, \tag{55}$$

where $H_t = \frac{2\theta}{L^2} \sum_{\tau=t_0+1}^{t} \alpha_\tau^2 \cdot (G_*^2 + 4L\Delta_\tau) + \frac{1}{M^2} \sum_{i,j \in [M]} \mathcal{B}_t^{i,j}$, and recall that $\theta := 48\eta^2 K^3 L^2$. Now, notice that $Q_t, H_t \geq 0$, and that $\theta$ satisfies $\theta K = 48\eta^2 K^4 L^2 \leq 1/2$ since we assume that $\eta^2 \leq \frac{1}{100 L^2 K^4}$ (see Eq. (8)). This enables to make use of Lemma 13 below to conclude,

$$
\begin{aligned}
Q_{t_0+1:t_0+K} &\leq 2H_{t_0+1:t_0+K} = \frac{4\theta}{L^2} \sum_{\tau=t_0+1}^{t_0+K} \alpha_\tau^2 \cdot (G_*^2 + 4L\Delta_\tau) + \frac{2}{M^2} \sum_{i,j \in [M]} \mathcal{B}_{t_0+1:t_0+K}^{i,j} \\
&= 200\eta^2 K^3 \sum_{\tau=t_0+1}^{t_0+K} \alpha_\tau^2 \cdot (G_*^2 + 4L\Delta_\tau) + \frac{2}{M^2} \sum_{i,j \in [M]} \mathcal{B}_{t_0+1:t_0+K}^{i,j} \;.
\end{aligned}
\tag{56}
$$

**Lemma 13.** *Let $K, \theta > 0$, and assume $\theta K \leq 1/2$. Also assume a sequence of non-negative terms $\{Q_t \geq 0\}_{t=t_0+1}^{t_0+K}$ and another sequence $\{H_t \geq 0\}_{t=t_0+1}^{t_0+K}$ that satisfy the following inequality for any $t \in [t_0, t_0 + K]$,*

$$Q_t \leq \theta Q_{t_0+1:t} + H_t$$

*Then the following holds,*

$$Q_{t_0+1:t_0+K} \leq 2H_{t_0+1:t_0+K} \;.$$

---

[8]Formally we should use the $i, j$ upper script for $\xi_\tau$ in the definition of $\mathcal{B}_t^{i,j}$, i.e. to define $\mathcal{B}_t^{i,j} := 8\eta^2 \alpha_t^2 \left\| \sum_{\tau=t_0+1}^{t} \alpha_{\tau:t} \frac{\alpha_\tau}{\alpha_t^2} \xi_\tau^{i,j} \right\|^2$. We absorb this notation into $\xi_\tau$ to simplify the notation.

Recalling that we like to bound the expectation of the LHS of Eq. (56), we will next bound $\left\|\sum_{\tau=t_0+1}^t \alpha_{\tau:t}\frac{\alpha_\tau}{\alpha_t^2}\xi_\tau\right\|^2$, which is done in the following Lemma [9],

**Lemma 14.** *The following bound holds for any $t \in [t_0, t_0 + K]$,*

$$\mathbf{E}\left\|\sum_{\tau=t_0+1}^t \frac{\alpha_{\tau:t}\alpha_\tau}{\alpha_t^2}\xi_\tau\right\|^2 \leq 4K^3\sigma^2 .$$

Since the above lemma for any $i,j$ and $t \in [t_0, t_0 + K]$ we can now bound $\mathbf{E}\mathcal{B}^{i,j}_{t_0+1:t_0+K}$ as follows,

$$
\begin{aligned}
\mathbf{E}\mathcal{B}^{i,j}_{t_0+1:t_0+K} &= 8\eta^2\alpha_t^2 \sum_{t=t_0+1}^{t_0+K} \mathbf{E}\left\|\sum_{\tau=t_0+1}^t \frac{\alpha_{\tau:t}\alpha_\tau}{\alpha_t^2}\xi_\tau\right\|^2 \\
&\leq 8\eta^2\alpha_t^2 \sum_{t=t_0+1}^{t_0+K} 4K^3\sigma^2 \\
&= 32\eta^2\alpha_t^2 K^4\sigma^2 \\
&= 32\eta^2(r+1)^2 K^6\sigma^2 ,
\end{aligned}
\tag{57}
$$

where the last lines $\alpha_t \leq (r+1)K$ for any iteration $t$ that belongs to round $r$.

Since the above holds for any $i, j$ it follows that,

$$\frac{2}{M^2}\sum_{i,j\in[M]} \mathcal{B}^{i,j}_{t_0+1:t_0+K} \leq 2\cdot 32\eta^2(r+1)^2 K^6\sigma^2 = 64\eta^2(r+1)^2 K^6\sigma^2 .$$

Plugging the above back into Eq. (56) gives,

$$Q_{t_0+1:t_0+K} \leq 200\eta^2 K^3 \sum_{\tau=t_0+1}^{t_0+K} \alpha_\tau^2\cdot(G_*^2 + 4L\Delta_\tau)) + 64\eta^2(r+1)^2 K^6\sigma^2 .\tag{58}$$

**Final Bound on $\mathbf{E}V_t$.** Finally, using the above bound together with the Eq. (46) enables to bound $\mathbf{E}V_t$ as follows,

$$
\begin{aligned}
\frac{1}{L^2}\mathbf{E}V_t &\leq \sum_{\tau=0}^t Q_\tau \\
&\leq \sum_{\tau=0}^T Q_\tau \\
&\leq 200\eta^2 K^3 \sum_{\tau=0}^T \alpha_\tau^2\cdot(G_*^2 + 4L\Delta_\tau)) + \sum_{r=0}^{R-1}\sum_{t=rK+1}^{rK+K} \alpha_t^2\mathbf{E}\|x_t^i - x_t^j\|^2 \\
&\leq 200\eta^2 K^3 \sum_{\tau=0}^T \alpha_\tau^2\cdot(G_*^2 + 4L\Delta_\tau)) + \sum_{r=0}^{R-1} 64\eta^2(r+1)^2 K^6\sigma^2 \\
&\leq 200\eta^2 K^3 \sum_{\tau=0}^T \alpha_\tau^2\cdot(G_*^2 + 4L\Delta_\tau)) + 64\eta^2 K^6\sigma^2\cdot\frac{8}{6}R^3 \\
&\leq 400\eta^2 K^3 \sum_{\tau=0}^T \alpha_{0:\tau}\cdot(G_*^2 + 4L\Delta_\tau)) + 90\eta^2 K^6 R^3\sigma^2 ,
\end{aligned}
$$

where we have used $\sum_{r=0}^{R-1}(r+1)^2 \leq \frac{8}{6}R^3$, and the last line uses $\alpha_\tau^2 = (\tau+1)^2 \leq 2\alpha_{0:\tau}$ Consequently, we can bound

$$\mathbf{E}V_t \leq 400L^2\eta^2 K^3 \sum_{\tau=0}^T \alpha_{0:\tau}\cdot(G_*^2 + 4L\Delta_\tau)) + 90L^2\eta^2 K^6 R^3\sigma^2 ,$$

which established the lemma.                                                           $\square$

---

[9]recall that for simplicity of notation we denote $\xi_\tau$ rather than $\xi_\tau^{i,j}$.

## J.1 Proof of Lemma 12

*Proof of Lemma 12.* First note that by definition of $x_\tau$ we have for any $i \in [M]$,

$$\|x_\tau - x_\tau^i\|^2 = \left\| \frac{1}{M} \sum_{l \in [M]} x_\tau^l - x_\tau^i \right\|^2 \leq \frac{1}{M} \sum_{l \in [M]} \|x_\tau^l - x_\tau^i\|^2 = \frac{1}{M\alpha_\tau^2} q_\tau^i . \tag{59}$$

where we have used Jensen's inequality, and the definition of $q_\tau^i$.

Using the above inequality, we obtain,

$$\begin{aligned}
\|\nabla f_i(x_\tau^i) - \nabla f_j(x_\tau^j)\|^2 &= \|(\nabla f_i(x_\tau^i) - \nabla f_i(x_\tau)) + (\nabla f_i(x_\tau) - \nabla f_j(x_\tau)) - (\nabla f_j(x_\tau^j) - \nabla f_j(x_\tau))\|^2 \\
&\leq 3\|\nabla f_i(x_\tau^i) - \nabla f_i(x_\tau)\|^2 + 3\|\nabla f_i(x_\tau) - \nabla f_j(x_\tau)\|^2 + 3\|\nabla f_j(x_\tau^j) - \nabla f_j(x_\tau)\|^2 \\
&\leq 3L^2 \left( \|x_\tau - x_\tau^i\|^2 + \|x_\tau - x_\tau^j\|^2 \right) + 6 \left( \|\nabla f_i(x_\tau)\|^2 + \|\nabla f_j(x_\tau)\|^2 \right) \\
&\leq \frac{3L^2}{M\alpha_\tau^2} (q_\tau^i + q_\tau^j) + 6 \left( \|\nabla f_i(x_\tau)\|^2 + \|\nabla f_j(x_\tau)\|^2 \right) ,
\end{aligned}$$

where the second and third lines use $\|\sum_{n=1}^N a_n\|^2 \leq N \sum_{n=1}^N \|a_n\|^2$ which holds for any $\{a_n \in \mathbb{R}^d\}_{n=1}^N$; we also used the smoothness of the $f_i(\cdot)$'s; and the last line uses Eq. (59). $\quad\square$

## J.2 Proof of Lemma 13

*Proof of Lemma 13.* Since the $Q_t$'s and $\theta$ are non-negative, we can further bound $Q_t$ for all $t \in [t_0, t_0 + K]$ as follows,

$$Q_t \leq \theta Q_{t_0+1:t} + H_t \leq \theta Q_{t_0+1:t_0+K} + H_t .$$

Summing the above over $t$ gives,

$$Q_{t_0+1:t_0+K} := \sum_{t=t_0+1}^{t_0+K} Q_t \leq \theta K \cdot Q_{t_0+1:t_0+K} + H_{t_0+1:t_0+K} \leq \frac{1}{2} \cdot Q_{t_0+1:t_0+K} + H_{t_0+1:t_0+K}$$

where we used $\theta K \leq 1/2$. Re-ordering the above equation immediately establishes the lemma. $\quad\square$

## J.3 Proof of Lemma 14

*Proof of Lemma 14.* Letting $\{\mathcal{F}_t\}_t$ be the natural filtration that is induces by the random draws up to time $t$, i.e., by $\{\{z_1^i\}_{i \in [M]}, \ldots, \{z_t^i\}_{i \in [M]}\}$. By the definition of $\xi_t$ it is clear that $\xi_t$ is measurable with respect to $\mathcal{F}_t$, and that,

$$\mathbf{E}[\xi_t | \mathcal{F}_{t-1}] = 0 .$$

Implying that $\{\xi_t\}_t$ is martingale difference sequence with respect to the filtration $\{\mathcal{F}_t\}_t$. The following implies that,

$$
\begin{aligned}
\mathbf{E} \left\| \sum_{\tau=t_0+1}^{t} \frac{\alpha_{\tau:t}\alpha_\tau}{\alpha_t^2} \xi_\tau \right\|^2 &\leq \sum_{\tau=t_0+1}^{t} \left( \frac{\alpha_{\tau:t}\alpha_\tau}{\alpha_t^2} \right)^2 \mathbf{E} \|\xi_\tau\|^2 \\
&\leq \sum_{\tau=t_0+1}^{t} \left( \frac{K\alpha_t \cdot \alpha_t}{\alpha_t^2} \right)^2 \mathbf{E} \|\xi_\tau\|^2 \\
&= K^2 \sum_{\tau=t_0+1}^{t} \mathbf{E} \|\xi_\tau\|^2 \\
&\leq K^2 \sum_{\tau=t_0+1}^{t} \mathbf{E} \left\| \xi_\tau^i - \xi_\tau^j \right\|^2 \\
&\leq 2K^2 \sum_{\tau=t_0+1}^{t} \left( \mathbf{E}\|\xi_\tau^i\|^2 + \mathbf{E}\|\xi_\tau^i\|^2 \right) \\
&\leq 2K^2 \sum_{\tau=t_0+1}^{t} 2\sigma^2 \\
&\leq 4K^2\sigma^2 \cdot (t - t_0) \\
&\leq 4K^3\sigma^2 ,
\end{aligned}
$$

where the first line follows by Lemma 15 below, the second line holds since $\alpha_\tau \leq \alpha_t$, and $\alpha_{\tau:t} \leq K\alpha_t$ (recall $\alpha_\tau \leq \alpha_t$ since $\tau \leq t$); the fourth line follows due to $\xi_\tau := \xi_\tau^i - \xi_\tau^j$; the fifth line uses $\|a+b\|^2 \leq 2\|a\|^2 + 2\|b\|^2$ for any $a, b \in \mathbb{R}^d$; the sixth line follows since $\mathbf{E}\|\xi_\tau^i\|^2 := \mathbf{E}\|g_\tau^i - \bar{g}_\tau^i\|^2 := \mathbf{E}\|\nabla f(x_\tau^i, z_\tau^i) - \nabla f(x_\tau^i)\|^2 \leq \sigma^2$; and the last line uses $(t - t_0) \leq K$.

**Lemma 15.** *Let $\{\xi_t\}_t$ be a martingale difference sequence with respect to a filtration $\{\mathcal{F}_t\}_t$, then the following holds for all time indexes $t_1, t_2 \geq 0$*

$$
\mathbf{E} \left\| \sum_{\tau=t_1}^{t_2} \xi_\tau \right\|^2 = \sum_{\tau=t_1}^{t_2} \mathbf{E} \|\xi_\tau\|^2 .
$$

$\square$

### J.3.1 Proof of Lemma 15

*Proof of Lemma 15.* We shall prove the lemma by induction over $t_2$. The base case where $t_2 = t_1$ clearly holds since it boils down to the following identity,

$$
\mathbf{E} \left\| \sum_{\tau=t_1}^{t_1} \xi_\tau \right\|^2 = \mathbf{E} \|\xi_{t_1}\|^2 = \sum_{\tau=t_1}^{t_1} \mathbf{E} \|\xi_\tau\|^2 .
$$

Now for induction step let us assume that the equality holds for $t_2 \geq t_1$ and lets prove it holds for $t_2 + 1$. Indeed,

$$
\begin{aligned}
\mathbf{E} \left\| \sum_{\tau=t_1}^{t_2+1} \xi_\tau \right\|^2 &= \mathbf{E} \left\| \xi_{t_2+1} + \sum_{\tau=t_1}^{t_2} \xi_\tau \right\|^2 \\
&= \mathbf{E} \left\| \sum_{\tau=t_1}^{t_2} \xi_\tau \right\|^2 + \mathbf{E} \|\xi_{t_2+1}\|^2 + 2\mathbf{E} \left( \sum_{\tau=t_1}^{t_2} \xi_\tau \right) \cdot \xi_{t_2+1} \\
&= \sum_{\tau=t_1}^{t_2+1} \mathbf{E} \|\xi_\tau\|^2 + 2\mathbf{E} \left[ \mathbf{E} \left[ \left( \sum_{\tau=t_1}^{t_2} \xi_\tau \right) \cdot \xi_{t_2+1} | \mathcal{F}_{t_2} \right] \right] \\
&= \sum_{\tau=t_1}^{t_2+1} \mathbf{E} \|\xi_\tau\|^2 + 2\mathbf{E} \left[ \left( \sum_{\tau=t_1}^{t_2} \xi_\tau \right) \cdot \mathbf{E} \left[ \xi_{t_2+1} | \mathcal{F}_{t_2} \right] \right] \\
&= \sum_{\tau=t_1}^{t_2+1} \mathbf{E} \|\xi_\tau\|^2 + 0 \\
&= \sum_{\tau=t_1}^{t_2+1} \mathbf{E} \|\xi_\tau\|^2 \ ,
\end{aligned}
$$

where the third line follows from the induction hypothesis, as well as from the law of total expectations; the fourth lines follows since $\{\xi_\tau\}_{\tau=0}^{t_2}$ are measurable w.r.t $\mathcal{F}_{t_2}$, and the fifth line follows since $\mathbf{E}[\xi_{t_2+1}|\mathcal{F}_{t_2}] = 0$. Thus, we have established the induction step and therefore the lemma holds. $\quad\square$

## K   Proof of Lemma 6

*Proof of Lemma 6.* Summing the inequality $A_t \leq \mathcal{B} + \frac{1}{2(T+1)} \sum_{t=0}^{T} A_t$ over $t$ gives,

$$
A_{0:T} \leq (T+1)\mathcal{B} + (T+1)\frac{1}{2(T+1)} A_{0:T} = (T+1)\mathcal{B} + \frac{1}{2} A_{0:T} \ ,
$$

Re-ordering we obtain,

$$
A_{0:T} \leq 2(T+1)\mathcal{B} \ .
$$

Plugging this back to the original inequality and taking $t = T$ gives,

$$
A_T \leq \mathcal{B} + \frac{1}{2(T+1)} A_{0:T} \leq 2\mathcal{B} \ .
$$

which concludes the proof. $\quad\square$

## L   The Necessity of Non-uniform Weights

One may wonder, why should we employ increasing weights $\alpha_t \propto t$ rather than using standard uniform weights $\alpha_t = 1 \ , \forall t$. Here we explain why uniform weights are insufficient and why increasing weights e.g. $\alpha_t \propto t$ are crucial to obtain our result.

**Intuitive Explanation.** Prior to providing an elaborate technical explanation we will provides some intuition. The intuition behind the importance of using increasing weights is the following: Increasing weights are a technical tool to put more emphasis on the last rounds. Now, in the context of Local update methods, the iterates of the last rounds are more attractive since the bias between different machines shrinks as we progress. Intuitively, this happens since as we progress with the optimization process, the bias in the stochastic gradients that we compute goes to zero (in expectation), and consequently the bias between different machines shrinks as we progress.

**Technical Explanation.** Assume general weights $\{\alpha_t\}_t$, and let us go back to the proof of Lemma 5 (see Section J). Recall that in this proof we derive a bound of the following form (see Eq. (55))

$$A_t \le \theta A_{t_0+1:t} + B_t \, , \tag{60}$$

where $A_t := \alpha_t^2 \|x_t^i - x_t^j\|^2$, $B_t = 8\eta^2 \alpha_t^2 \left\| \sum_{\tau=t_0+1}^t \alpha_{\tau:t} \frac{\alpha_\tau}{\alpha_t^2} \xi_\tau \right\|^2$, and importantly [10],

$$\theta := \eta^2 \frac{2\alpha_t^4}{(\alpha_{0:t})^2} \cdot K^3 L^2 \, .$$

Now, a crucial part of the proof is the fact that $\theta K \le 1/2$, which in turn enables to employ Lemma 13 in order to bound $\mathbf{E} V_t$.

Not let us inspect the constraint $\theta K \le 1/2$ for polynomial weights of the form $\alpha_t \propto t^p$ where $p \ge 0$. This condition boils down to,

$$\eta^2 \frac{2\alpha_t^4}{(\alpha_{0:t})^2} \cdot K^3 L^2 \cdot K \le 1/2 \, ,$$

Implying the following bound should apply to $\eta$ *for any* $t \in [T]$,

$$\eta \le \frac{1}{2LK^2} \cdot \frac{\alpha_{0:t}}{\alpha_t^2} \approx \frac{1}{2LK^2} \cdot \frac{t^{p+1}}{t^{2p}} = \frac{1}{2LK^2} \cdot t^{1-p}$$

Now since the bound should hold for any $t \in [T]$ we could divide into two cases:
**Case 1:** $p \le 1$. In this case $t^{1-p}$ is monotonically increasing with $t$ so the above condition should be satisfied for the smallest $t$, namely $t = 1$, implying,

$$\eta \le \frac{1}{2LK^2} \cdot \frac{\alpha_{0:t}}{\alpha_t^2} \approx \frac{1}{2LK^2} \, .$$

The effect of this term on the overall error stems from the first term in the error analysis (see e.g. Eq. (36)), namely,

$$\frac{1}{\alpha_{0:T}} \cdot \frac{\|w_0 - w^*\|^2}{\eta} = \frac{2LK^2 \cdot \|w_0 - w^*\|^2}{T^{p+1}}$$

Now, for the extreme values $p = 0$ (uniform weights) and $p = 1$ (linear weights), the above expression results an error term of the following form,

$$\text{Err}(\mathbf{p} = \mathbf{0}) = O\left(K^2/T\right) = O(K/R) \quad \& \quad \text{Err}(\mathbf{p} = \mathbf{1}) = O(K^2/T^2) = O(1/R^2) \, . \tag{61}$$

Thus, for $p = 0$, the error term is considerably worse even compared to Minibatch-SGD, and $p = 1$ is clearly an improvement.
**Case 2:** $p > 1$. In this case $t^{1-p}$ is decreasing increasing with $t$ so the above condition should be satisfied for the largest $t$, namely $t = T$, implying,

$$\eta \le \frac{1}{2LK^2} \cdot \frac{\alpha_{0:t}}{\alpha_t^2} \approx \frac{1}{2LK^2} \cdot \frac{1}{T^{p-1}} \, .$$

Now, the effect of this term on the overall error stems from the first term in the error analysis (see e.g. Eq. (36)), namely,

$$\frac{1}{\alpha_{0:T}} \cdot \frac{\|w_0 - w^*\|^2}{\eta} = \frac{2LK^2 \cdot \|w_0 - w^*\|^2 \cdot T^{p-1}}{T^{p+1}} = \frac{2LK^2 \cdot \|w_0 - w^*\|^2 \cdot}{T^2}$$

Thus, for any $p \ge 1$ we obtain,

$$\text{Err}(\mathbf{p}) = O(K^2/T^2) = O(1/R^2) \, . \tag{62}$$

**Conclusion:** As can be seen from Equations (61) (62), using uniform weights or even polynomial weights $\alpha_t \propto t^p$ with $p < 1$ yields strictly worse guarantees compared to taking $p \ge 1$.

---

[10] Indeed, in the case where $\alpha_t := t + 1$ we can take $\theta := 8\eta^2 K^3 L^2$, and this is used in the proof of Lemma 5

# M Experiments

## M.1 Data Distribution Across Workers

In federated learning or distributed machine learning settings, data heterogeneity among workers often arises due to non-identical data distributions. To simulate such scenarios, we split the MNIST dataset using a Dirichlet distribution. The Dirichlet distribution allows control over the degree of heterogeneity in the data assigned to each worker by adjusting the *dirichlet-alpha* parameter. Lower values of *dirichlet-alpha* result in more uneven class distributions across workers, simulating highly non-IID data.

Given $C$ classes and $M$ workers, the probability $p_{c,m}$ of assigning data from class $c$ to worker $m$ is based on sampling from a Dirichlet distribution:

$$p_{c,1}, p_{c,2}, \ldots, p_{c,M} \sim \text{Dirichlet}(\alpha)$$

where $\alpha$ is the concentration parameter controlling the level of heterogeneity. A smaller $\alpha$ value results in a more imbalanced distribution, meaning that each worker primarily receives data from a limited subset of classes. In this experiment, we set $\alpha = 0.1$ to induce high heterogeneity.

The following figures present scatter plots illustrating the class frequencies assigned to individual workers for different worker configurations—16, 32, and 64—using a specific random seed.

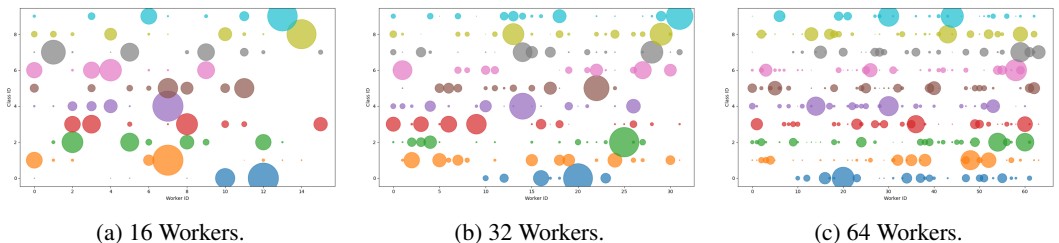

      (a) 16 Workers.            (b) 32 Workers.           (c) 64 Workers.

Figure 2: Class distribution across workers for different numbers of workers (16, 32, and 64) on the MNIST dataset. The dataset was partitioned using a Dirichlet distribution, with the *dirichlet-alpha* parameter set to 0.1 to induce high heterogeneity. Each scatter plot illustrates class frequencies for each worker.

## M.2 Complete Experimental Results

This section presents the complete experimental results for 16, 32, and 64 workers, showing test accuracy and test loss as functions of local iterations $K$. The plots illustrate the method's scalability and performance, with higher accuracy ($\uparrow$) and lower loss ($\downarrow$) indicating better outcomes.

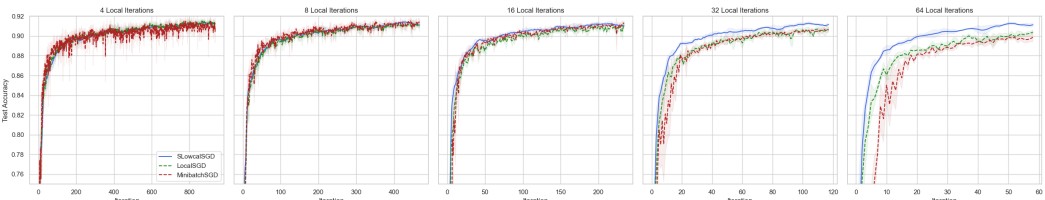

Figure 3: Test Accuracy vs. Local Iterations ($K$) for 16 workers ($\uparrow$ is better).

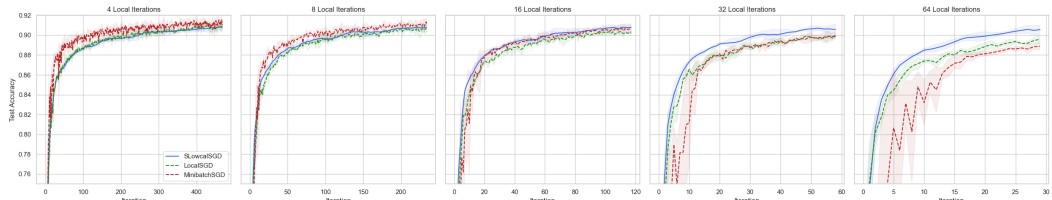

Figure 4: Test Accuracy vs. Local Iterations ($K$) for 32 workers ($\uparrow$ is better).

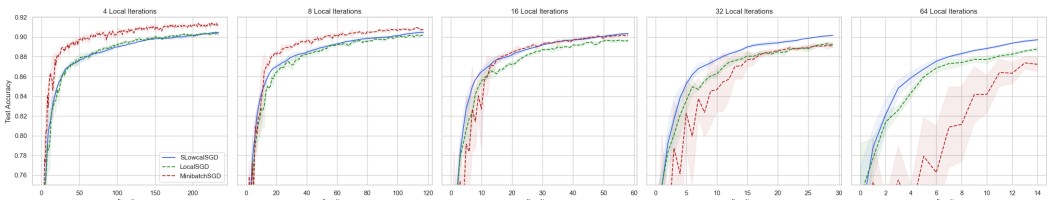

Figure 5: Test Accuracy vs. Local Iterations ($K$) for 64 workers ($\uparrow$ is better).

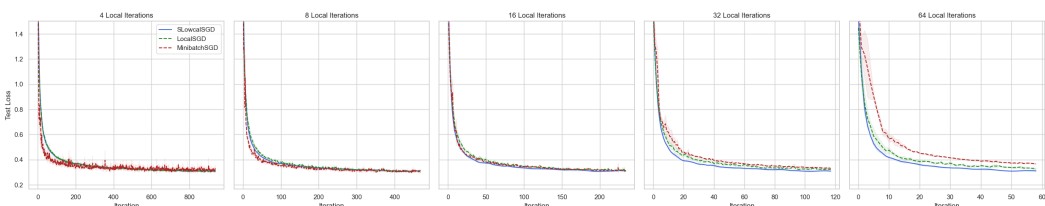

Figure 6: Test Loss vs. Local Iterations ($K$) for 16 workers ($\downarrow$ is better).

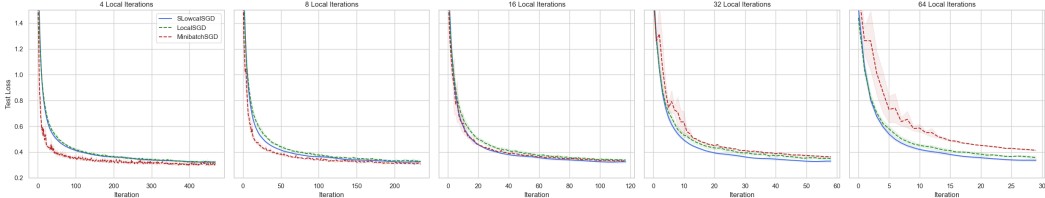

Figure 7: Test Loss vs. Local Iterations ($K$) for 32 workers ($\downarrow$ is better).

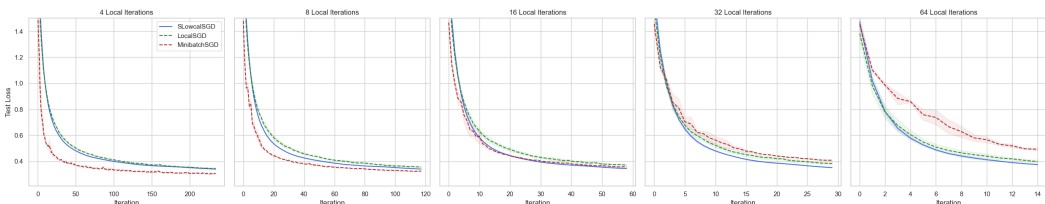

Figure 8: Test Loss vs. Local Iterations ($K$) for 64 workers ($\downarrow$ is better).

