# OpenReview forum: "SLowcalSGD : Slow Query Points Improve Local-SGD for Stochastic Convex Optimization"
_NeurIPS.cc/2024/Conference — NeurIPS 2024 poster_

### Official Review · Reviewer_mxJX · 2024-07-15

**Soundness:** 4
**Presentation:** 4
**Contribution:** 4
**Rating:** 7
**Confidence:** 4

**Summary:**

This work studies distributed model training with a parameter server; contributions are theoretical:
* **Assumptions**: $L$-smooth convex objectives, stochastic gradient estimates, heterogeneous worker distributions, bounded difference in expected gradients of local workers' objectives at a global minima, denoted $G^\star$.
* **Main result**: Given a fixed set of communication rounds $R$ with the parameter server, $M$ worker machines, and a budget of $K$ local gradient queries per worker in each communication round, this work presents a simple method for improving local SGD. Given $L$-smooth convex objectives with stochastic gradients and heterogeneous worker distributions, the local-SGD variant proposed in this work achieves more favorable convergence than mini-batch SGD (where the $K$ stochastic gradient queries are used to compute a larger mini-batch gradient at each worker). The proposed method converges in $\mathcal{O}(MK^{-1/2})$ rounds, whereas mini-batch SGD converges in $\mathcal{O}(MK)$ rounds.
* **Method**: The proposed method extends anytime GD (Cutkosky, ICML'19) to the distributed local-sgd parameter-server setting. In short, regular local-sgd with $K$ local steps, but where stochastic gradients are computed at an exponential moving average (ema) of the model parameters. The parameter server averages both the primal and ema parameters, and sends back the exact average to each worker at the end of each round.

**Strengths:**

**Clarity**
* This work is extremely clear and well written. I have skimmed the main proofs in the appendix, but the proof sketches in the main paper provide enough detail to understand and follow the primary logic and intuition behind each result.

**Quality**
* I have quickly gone over the proofs in the appendix, and the work appears to be technically sound.

**Originality**
* To the best of my knowledge, this is the first such extension of anytime GD to the distributed stochastic local update setting, and the first result demonstrating the improvements of local sgd over mini-batch sgd in general smooth convex settings. Previous studies were either devoted to the non-convex setting or quadratic objectives, or assumed bounded second moments.

**Weaknesses:**

**Significance**
* No numerical experiments conducted to explore the convergence of the proposed method or generalization capabilities. To increase impact and adoption, researchers may be interested in the generalization performance of the method; i.e., going beyond the number of communication rounds required to decrease training error. Would design considerations (e.g., choice of $\alpha_t$ schedule) change in such a setting?
* Not a major issue for a theoretical paper, but current hyper-parameter choices requires knowledge of problem parameters ($\sigma$ and L-smoothness constant).

**Clarity**
* Minor complaint is that the proof sketches do not always correspond to a similar logic used in the actual proofs themselves. For instance, (16) in the proof sketch in the main paper was obtained by assuming a somewhat monotonic iterate sequence, but such a result is never explicitly proven in the appendix. Instead, Lemma 3 in Appendix H (which bounds the sum of the iterate sequence using the T times the starting error, plus additional terms depending on the gradient magnitude and variance; which should indeed converge) is used to arrive at (33) in the proof of Theorem 2 in the appendix, which corresponds to (16) in the main paper.

**Questions:**

* Please consider including simple numerical experiments in the convex setting (e.g., multinomial logistic regression or least squares) to empirically evaluate the performance of the proposed method.
* Would appreciate a discussion on the main challenges to demonstrating improved convergence relative to accelerated mini-batch SGD (not just vanilla mini-batch SGD), especially since the proposed EMA update can be given a momentum-like interpretation as stated in Appendix C.

**Limitations:**

N.A.

---

> ### Author Rebuttal · Authors · 2024-08-06
>
> **Response to Reviewer mxJX**
>
>
> Thank you for your supportive review, below we address the points that you have raised.
>
>
> **Regarding weaknesses**
>
>
> **Q:** adding experiments
>
>
> **A:** We have added experiments that demonstrate the benefit of our approach and corroborate our theoretical findings. Please see the details in our response to all reviewers.
>
>
> **Q:** “...researchers may be interested in the generalization performance of the method…Would design considerations (e.g., choice of schedule) change in such a setting?”
>
>
> **A:** Thank you for this comment. In fact we do derive guarantees for generalization (expected excess loss).
> This can be directly seen from our description of the problem (see Section 2 lines 81-97, and specifically lines 92-94), as well as from the guarantees that we state in Theorem 2 (lines 191-196). In our experiments we present both test loss and test error.
>
>
> **Q:** “not a major issue but…..learning rate requires knowledge of problem parameters”
>
>
> **A:** This is a good point. Our focus in the paper was on showing that one can improve over the minibatch-sgd baseline in the heterogeneous case, which was already quite challenging. We agree that developing and exploring adaptive (or parameter free) methods for local training methods is an important topic, and we hope to extend our work in this respect in the future. It is interesting to note that (as far as we know) there does not exist an adaptive (or parameter free) variant of the standard local-sgd baseline, which is in itself an attractive future direction.
>
>
> **Q:** “Minor complaint … proof sketches do not always correspond to a similar logic used in the actual proofs themselves”
>
>
> **A:** Indeed, as the reviewer noticed we made a simplification in the proof sketch, and the goal was to simplify the presentation of ideas and uncover the main ideas and intuition. Please note, that when we simplify the proof sketch we do mention this explicitly: in line 260 just before equation (16) we write “to simplify the proof sketch we shall assume that $D_t \leq D_0$”.
> In the final version we shall mention explicitly that this simplification does not hold in the full proof.
>
>
>
>
>
>
>
>
>
>
> **Regarding questions**
>
>
> **Q:** “a discussion on the main challenges to demonstrating improved convergence relative to accelerated mini-batch SGD”
>
>
> **A:** Thank you for raising this important point. In order to improve over accelerated minibatch-sgd, we have considered designing an accelerated variant of our approach. Nevertheless, there are several challenges in doing so: First, one can think of incorporating an “acceleration ingredient" to both aggregated and local steps which inserts another degree of freedom to the design of the algorithm, and it is not very clear what is the right way to do so. Second, accelerated methods are more complicated to analyze. Finally, it is not clear how acceleration can better mitigate the bias between different machines, which is a main obstacle in improving over our current approach.
>
>
> Ideally, we believe that one can find a way to optimally tradeoff between acceleration and bias (by controlling the learning rate and weighting scheme), thus leading towards better guarantees.
> We will add this discussion to the final version of the paper.

---

### Official Review · Reviewer_5jSD · 2024-07-17

**Soundness:** 3
**Presentation:** 4
**Contribution:** 3
**Rating:** 6
**Confidence:** 4

**Summary:**

This paper introduces a new federated learning algorithm called Slowcal-SGD, which basically introduces anytime-SGD into federated learning setting. The authors provide solid convergence analysis and fruitful insights on the new algorithm. They show the algorithm can provably beat both mini-batch SGD and local SGD in the heterogeneous data setting. No experiments are provided, though.

**Strengths:**

- For each theorem, the authors provided insightful discussions, explaining why the algorithm works better.
- The proposed algorithm is novel and neat.

**Weaknesses:**

- It'd be better to define "query". It could be controversial. For example, why is computing the gradients wrt a mini-batch a single query instead of B (batch size) queries?
- No experimental results are provided. So it's hard to tell whether the proposed algorithm works in practice.

**Questions:**

See above comments

---

> ### Author Rebuttal · Authors · 2024-08-06
>
> **Response to Reviewer 5jSD**
>
> Thank you for your supportive review, below we address the points that you have raised.
>
>
> **Regarding weaknesses**
>
>
> **Q:** adding experiments
>
>
> **A:** We have added experiments that demonstrate the benefit of our approach and corroborate our theoretical findings. Please see the details in our response to all reviewers.
>  Since the experiments were your main concern, we will appreciate it if you can consider raising your score given the experiments we conducted.
>
>
> **Q:** definition of “query”
>
>
>  **A:**  When we use the term “query point” we refer to the point (model parameter) at which we estimate the gradient (of the expected loss).
>
>
>
> **Q:** “..., why is computing the gradients wrt a mini-batch a single query instead of B (batch size) queries?”
>
> **A:**  in your question when you use the word “query” you mean the number of samples (or stochastic gradient computations) that are employed. We do not use the term “query” but rather the term samples or number of (stochastic) gradient computations. This is since we already use the tem “query point” (as we explain above) and we do not want to confuse it with the word “query”. Specifically, when we relate to a batchsize of $B$ we indeed count it as $B$ samples and $B$ (stochastic) gradient computations.
> We will clarify this in the final version of the paper.

---

### Official Review · Reviewer_3Xai · 2024-07-17

**Soundness:** 2
**Presentation:** 2
**Contribution:** 2
**Rating:** 4
**Confidence:** 4

**Summary:**

This paper proposes SLowcal-SGD, which is a distributed learning algorithm that builds on customizing a recent technique for incorporating a slowly-changing sequence of query points, which in turn enables to better mitigate the bias induced by the local updates. Theoretical proof is given on the proposed algorithms.

**Strengths:**

* The idea of combining local updates and mini-batch SGD to improve the data heterogeneity case is interesting.
* The author provides intuition on why slowcal-SGD is useful, and the comparison of proposed algorithm and other algorithms in table 1 is clear.

**Weaknesses:**

* While the paper is heavy on theory, it has no validation on any synthetic or real-world data. It should not be hard to verify this since this is convex problems and there are multiple ways and open sourced code to produce heterogeneous data.
* The rates shown in table 1 is a little confusing. Comparing to accelerated mini-batch SGD, what is the advantage of proposed slowcal-SGD?
* The assumptions of equation (1) - (3) are somewhat strong if we discuss the data heterogeneity. What is the main difference between proposed slowcal-SGD and general variance reduction SGD? (which doesn't need equation 1-3).

**Questions:**

Please refer to the previous section.

**Limitations:**

The paper is theory-driven and does not have negative societal impact.

---

> ### Author Rebuttal · Authors · 2024-08-06
>
> **Response to Reviewer 3Xai**
>
> Thank you for your comments, we address your concerns below and kindly ask you to raise your score accordingly.
>
>
> **Regarding weaknesses**
>
>
>
> **Q:** adding experiments
>
> **A:** We have added experiments that demonstrate the benefit of our approach and corroborate our theoretical findings. Please see the details in our response to all reviewers.
>
>
>
>
> **Q:** comparison to accelerated minibatch sgd in table 1
>
> **A:** Indeed, as we describe in our paper, currently there does not exist a baseline that improves over the accelerated-minibatch-sgd, and this also applies even for the simpler homogeneous case.
> In the heterogeneous case, no method prior to our work was able to improve over the simpler minibatch-sgd baseline, and we are the first to establish such guarantees. We hope that the new technique that we introduced in our paper may pave towards designing an approach that will be able to improve over the  accelerated-minibatch-sgd baseline.
>
>
>
>
> **Q:** Assumption on heterogeneity
>
> **A:** Assumption in Equations (1-3) are indeed related to heterogeneity. But please note that we only consider the Assumption in Equation (1) to hold, which is the less restrictive assumption (Equation (2) is a consequence of Assumption (1), and Equation (3) is a much stronger assumption than (1) ).
>
> The heterogeneity assumption is not required for parallel training methods like minibatch-sgd and its accelerated version, since in such methods the query points of all workers are fully synchronized. Nevertheless, in local update methods (like Local-sgd and SLowcal-sgd) there is a drift in the query points of different workers (due to the local updates) and therefore heterogeneity must comes into play (this is also evident from existing lower bound for local-sgd).

---

### Author Rebuttal · Authors · 2024-08-07

**Dear reviewers**,

We have now added experimental results comparing our approach to several baselines. Our results showcase the practicality and benefit of our approach and complement our theoretical findings.

We have conducted  experiments on the MNIST dataset, which is a widely used benchmark in machine learning. The dataset consists of 70,000 grayscale images of handwritten digits (0-9), with 60,000 images in the training set and 10,000 images in the test set. We have executed our experiments on the NVIDIA GeForce RTX 3090 GPU and the PyTorch framework.

We employed a logistic regression model and compared our SLowcalSGD with LocalSGD and MinibatchSGD across various configurations. Specifically, we tested with 16, 32, and 64 workers and varied the number of local steps (or minibatch sizes for MinibatchSGD) to 1, 4, 8, 16, 32, and 64. For each local update in the SLowcalSGD and LocalSGD, we used a single sample, with the weights for SLowcalSGD set as \(\alpha_t = t\). We used a learning rate of 0.01, optimized through grid search. To ensure the reliability of our results, we conducted our experiments using three different random seeds and reported the average results across these seeds. We made one pass on the MNIST dataset for all experiments to ensure a fair comparison.

Our results indicate that as the number of local steps K (or minibatch sizes for MinibatchSGD) increases, SLowcalSGD exhibits more significant advantages over LocalSGD. Notably, both approaches, SLowcalSGD and LocalSGD achieve better performance compared to Minbatch-SGD.

---

### Decision · Program_Chairs · 2024-09-25

**Decision:**

Accept (poster)

**Comment:**

This paper gives a convergence guarantee for a distributed learning algorithm that improves upon both baselines of mini-batch SGD and local SGD. It does so by leveraging a slowly-changing sequence of query points (see Anytime-SGD), which in turn enables to better mitigate the client drift induced by the local updates. The paper provides novel insights, is well executed and interesting, as acknowledged by the reviews.

For the camera-ready version, we urge the authors to incorporate the mentioned feedback by the reviewers.